# Differential protection against SARS-CoV-2 reinfection pre- and post-Omicron

Hiam Chemaitelly[1,2,3 ✉], Houssein H. Ayoub[4], Peter Coyle[5,6,7], Patrick Tang[8], Mohammad R. Hasan[9], Hadi M. Yassine[5,10], Asmaa A. Al Thani[5,10], Zaina Al-Kanaani[6], Einas Al-Kuwari[6], Andrew Jeremijenko[6], Anvar Hassan Kaleeckal[6], Ali Nizar Latif[6], Riyazuddin Mohammad Shaik[6], Hanan F. Abdul-Rahim[11], Gheyath K. Nasrallah[5,10], Mohamed Ghaith Al-Kuwari[12], Adeel A. Butt[3,6,13], Hamad Eid Al-Romaihi[14], Mohamed H. Al-Thani[14], Abdullatif Al-Khal[6], Roberto Bertollini[14] & Laith J. Abu-Raddad[1,2,3,11,15 ✉]

The severe acute respiratory syndrome coronavirus 2 (SARS-CoV-2) has rapidly evolved over short timescales, leading to the emergence of more transmissible variants such as Alpha and Delta[1–3]. The arrival of the Omicron variant marked a major shift, introducing numerous extra mutations in the spike gene compared with earlier variants[1,2]. These evolutionary changes have raised concerns regarding their potential impact on immune evasion, disease severity and the effectiveness of vaccines and treatments[1,3]. In this epidemiological study, we identified two distinct patterns in the protective effect of natural infection against reinfection in the Omicron versus pre-Omicron eras. Before Omicron, natural infection provided strong and durable protection against reinfection, with minimal waning over time. However, during the Omicron era, protection was robust only for those recently infected, declining rapidly over time and diminishing within a year. These results demonstrate that SARS-CoV-2 immune protection is shaped by a dynamic interaction between host immunity and viral evolution, leading to contrasting reinfection patterns before and after Omicron's first wave. This shift in patterns suggests a change in evolutionary pressures, with intrinsic transmissibility driving adaptation pre-Omicron and immune escape becoming dominant post-Omicron, underscoring the need for periodic vaccine updates to sustain immunity.

Contrary to an initial notion of slow evolution[1], the severe acute respiratory syndrome coronavirus 2 (SARS-CoV-2) has demonstrated rapid evolutionary changes over short timescales[1–3], consistent with those of coronaviruses and RNA viruses in general[1]. In the pre-Omicron era, various divergent lineages of SARS-CoV-2 emerged, giving rise to distinct variants such as Alpha, Beta and Delta[1], each with its specific phenotypic traits[1,4–6].

However, the emergence of the Omicron variant in late 2021 marked a major shift, as it harboured dozens of further mutations in the spike gene compared with its predecessors[1,2]. Since then, the Omicron lineage has continued to evolve, leading to the emergence of new variants[1,2]. These rapid evolutionary changes have sparked concerns regarding their potential implications for immune evasion, transmissibility, disease severity, diagnostic accuracy and the effectiveness of existing vaccines and treatments[1,3].

SARS-CoV-2 infection has been shown to provide protective effects against reinfection under various scenarios, such as during the dominance of different strains[7–9] and against specific variants[10–13]. Studies have also investigated the waning of immune protection conferred by pre-Omicron infection against both pre-Omicron and Omicron reinfections[7–9,14]. However, investigations into the durability of this protection have been limited with existing studies focusing on relatively short timeframes because of the recentness of the pandemic. Notably, the durability of Omicron infection in preventing reinfection with Omicron itself over extended periods of time remains unknown.

In this study, our objective was to examine the consequences of viral evolution, particularly the transition from the pre-Omicron era to the Omicron era, on the level and durability of protection provided by natural immunity. Natural immunity refers here to the protection gained from a previous infection against reinfection and against severe,

[1]Infectious Disease Epidemiology Group, Weill Cornell Medicine–Qatar, Cornell University, Qatar Foundation – Education City, Doha, Qatar. [2]World Health Organization Collaborating Centre for Disease Epidemiology Analytics on HIV/AIDS, Sexually Transmitted Infections, and Viral Hepatitis, Weill Cornell Medicine–Qatar, Cornell University, Qatar Foundation – Education City, Doha, Qatar. [3]Department of Population Health Sciences, Weill Cornell Medicine, Cornell University, New York, NY, USA. [4]Mathematics Program, Department of Mathematics, Statistics and Physics, College of Arts and Sciences, Qatar University, Doha, Qatar. [5]Department of Biomedical Science, College of Health Sciences, QU Health, Qatar University, Doha, Qatar. [6]Hamad Medical Corporation, Doha, Qatar. [7]Wellcome-Wolfson Institute for Experimental Medicine, Queens University, Belfast, UK. [8]Department of Pathology, Sidra Medicine, Doha, Qatar. [9]Department of Pathology and Molecular Medicine, McMaster University, Hamilton, Ontario, Canada. [10]Biomedical Research Center, QU Health, Qatar University, Doha, Qatar. [11]Department of Public Health, College of Health Sciences, QU Health, Qatar University, Doha, Qatar. [12]Primary Health Care Corporation, Doha, Qatar. [13]Department of Medicine, Weill Cornell Medicine, Cornell University, New York, NY, USA. [14]Ministry of Public Health, Doha, Qatar. [15]College of Health and Life Sciences, Hamad bin Khalifa University, Doha, Qatar. ✉e-mail: hsc2001@qatar-med.cornell.edu; lja2002@qatar-med.cornell.edu

critical or fatal coronavirus disease 2019 (COVID-19) on reinfection. We specifically investigated the level and durability of the effectiveness of Omicron infection in preventing reinfection with an Omicron virus, contrasting it with the effectiveness of a pre-Omicron infection in preventing reinfection with a pre-Omicron virus.

The effectiveness of natural infection against reinfection was estimated in Qatar's population, both overall and by time since previous infection, using the test-negative, case-control study design[11,12,15,16]. Cases (SARS-CoV-2-positive tests) and controls (SARS-CoV-2-negative tests) were matched exactly one-to-two by sex, 10-year age group, nationality, number of coexisting conditions, number of vaccine doses, calendar week of the SARS-CoV-2 test, method of testing (polymerase chain reaction (PCR) or rapid antigen) and reason for testing, to balance observed confounders that could influence the risk of infection across the exposure groups[17].

Whereas evidence on viral evolution and immunity at the molecular level indicates the continuing emergence of new SARS-CoV-2 variants with resistance to neutralization by plasma from recovered individuals and sera from vaccinated individuals[1,2,18–20], the implications of these findings for the population-level phenomenon of immune protection remain uncertain. The present study provides direct population-level evidence contrasting the functional impact of viral evolution on immune protection in the pre-Omicron versus Omicron eras.

## Study populations

Extended Data Figures 1 and 2 illustrate the process of selecting the study populations for estimating the effectiveness of a pre-Omicron infection in preventing reinfection with a pre-Omicron virus and of an Omicron infection in preventing reinfection with an Omicron virus, respectively. Extended Data Figure 3 describes infection incidence at times of dominance of different SARS-CoV-2 variants in Qatar.

Extended Data Table 1 shows the characteristics of the study populations. The study was carried out on Qatar's entire population; therefore, the study population is representative of the internationally diverse but predominantly young and male demographic of the country.

## Immune protection in the pre-Omicron era

The overall effectiveness of a pre-Omicron infection in preventing reinfection, regardless of symptoms, with a pre-Omicron virus was estimated at 81.1% (95% confidence interval (CI), 80.4–81.8%) (Table 1a and Fig. 1a). The median duration between the previous infection and the study SARS-CoV-2 test was 252 days (interquartile range (IQR), 175–313 days). This robust protection showed limited waning over time after the previous infection. Effectiveness was 81.3% (95% CI, 80.6–82.1%) in the first year after the previous infection and 79.5% (95% CI, 77.1–81.5%) thereafter.

The effectiveness of a pre-Omicron infection in preventing symptomatic reinfection with a pre-Omicron virus demonstrated a similar pattern to that observed for any reinfection (Table 2a). The overall effectiveness against symptomatic reinfection was 86.8% (95% CI, 85.7–87.9%), with no evidence of waning over time after the previous infection. The median duration between the previous infection and the study SARS-CoV-2 test was 244 days (IQR, 169–303 days).

Subgroup analyses for both unvaccinated and vaccinated individuals yielded results similar to those of the main analysis (Table 3a and Extended Data Fig. 4a). The sensitivity analysis using a 40-day window[21] for defining reinfection instead of the 90-day window also produced results consistent with the main analysis (Extended Data Table 2a).

The overall effectiveness of a pre-Omicron infection in preventing severe, critical or fatal COVID-19 on reinfection with a pre-Omicron virus was 98.0% (95% CI, 96.1–99.0%), with no observed waning over time after the previous infection (Table 1b and Fig. 1b).

## Immune protection in the Omicron era

The overall effectiveness of an Omicron infection in preventing reinfection, regardless of symptoms, with an Omicron virus was estimated at 53.6% (95% CI, 52.1–55.0%) (Table 1c and Fig. 1a). The median duration between the previous infection and the study SARS-CoV-2 test was 245 days (IQR, 191–311 days).

This effectiveness demonstrated a rapid decline over time after the previous infection, decreasing from 81.3% (95% CI, 79.6–82.9%) within 3 to less than 6 months after the previous infection to 59.8% (95% CI, 57.8–61.7%) in the subsequent 3 months, and further dropping to 27.5% (95% CI, 22.7–32.0%) in the subsequent 3 months (Table 1c and Fig. 1a). Ultimately, it reached a negligible level after 1 year. The effectiveness was 59.5% (95% CI, 58.0–60.9%) in the first year after the previous infection and 4.8% (95% CI, −2.7–11.8%) thereafter.

The effectiveness of an Omicron infection in preventing symptomatic reinfection with an Omicron virus demonstrated a similar pattern to that observed for any reinfection (Table 2b). The overall effectiveness against symptomatic reinfection was 45.4% (95% CI, 42.5–48.2%), with a rapid decline observed over time after the previous infection. The median duration between the previous infection and the study SARS-CoV-2 test was 301 days (IQR, 225–457 days).

Subgroup analyses for both unvaccinated and vaccinated individuals yielded results similar to those of the main analysis (Table 3b and Extended Data Fig. 4b). The sensitivity analysis using a 40-day window[21] for defining reinfection instead of the 90-day window also produced results consistent with the main analysis (Extended Data Table 2b).

The overall effectiveness of an Omicron infection in preventing severe, critical or fatal COVID-19 on reinfection with an Omicron virus was 100% (95% CI, 79.9–100%), with no cases of reinfection progressing to severe, critical or fatal COVID-19 (Table 1d). There was no evidence for a waning in this effectiveness over time after the previous infection (Fig. 1b).

## Further validation analyses

### Previous infection misclassification

Under-ascertainment of infection leads to misclassification of previous infection status in the test-negative design used in this study, potentially biasing the estimates[15]. Extended Data Figure 5 presents the results of mathematical modelling simulations that evaluated the impact of extreme under-ascertainment (90% of SARS-CoV-2 infections are undocumented) on the estimated waning pattern of immune protection during the pre-Omicron and Omicron eras.

The estimated effectiveness in presence of this bias, in both pre-Omicron and Omicron eras, was comparable to, but slightly lower than, the true effectiveness (Extended Data Fig. 5). This underestimation was larger in the Omicron analysis. However, in both analyses, this bias did not affect the estimated duration of immune protection, which is the central focus of our study.

### Coexisting conditions misclassification

Coexisting conditions were identified by analysing electronic health record encounters within the national public healthcare system's database (Supplementary Methods section 1). However, this approach may not capture all conditions, as some may be undiagnosed or diagnosed at private facilities.

The sensitivity analysis, which removed matching by the number of coexisting conditions to simulate a scenario of complete under-ascertainment, showed results nearly identical to the main analysis across the various estimates (Extended Data Figs. 6 and 7). This consistency was observed for both the pre-Omicron and Omicron eras, including overall effectiveness against any infection and against infection in

**Table 1 | Effectiveness of previous infection against reinfection**

| Effectiveness | Cases[a] | | Controls[a] | | Effectiveness[b] (%) (95% CI)[c] | Cases[d] | | Controls[d] | | Effectiveness[b] (%) (95% CI)[c] |
|---|---|---|---|---|---|---|---|---|---|---|
| | Previous infection (n) | No previous infection (n) | Previous infection (n) | No previous infection (n) | | Previous infection (n) | No previous infection (n) | Previous infection (n) | No previous infection (n) | |
| **Effectiveness of a pre-Omicron infection in preventing** | | | | | | | | | | |
| **(a) Reinfection with a pre-Omicron virus** | | | | | | **(b) Severe, critical or fatal COVID-19 on reinfection with a pre-Omicron virus** | | | | |
| Any previous infection | 2,973 | 381,258 | 26,880 | 657,377 | 81.1 (80.4 to 81.8) | 9 | 9,824 | 1,505 | 36,193 | 98.0 (96.1 to 99.0) |
| By time since previous infection | | | | | | | | | | |
| Subgroup analysis 1 | | | | | | | | | | |
| 3–<6 months | 870 | 378,591 | 7,018 | 656,824 | 77.9 (76.3 to 79.5) | 2 | 9,762 | 355 | 36,191 | 98.1 (92.5 to 99.5) |
| 6–<9 months | 686 | 378,980 | 8,330 | 656,893 | 85.8 (84.6 to 86.9) | 3 | 9,771 | 463 | 36,190 | 97.8 (93.2 to 99.3) |
| 9–<1 year | 1,007 | 379,405 | 8,124 | 656,972 | 79.7 (78.3 to 81.0) | 4 | 9,776 | 615 | 36,190 | 97.8 (94.1 to 99.2) |
| ≥1 year | 389 | 378,150 | 3,385 | 656,760 | 79.5 (77.1 to 81.5) | 0 | 9,745 | 72 | 36,188 | 100.0 (94.7 to 100.0)[e] |
| Subgroup analysis 2 | | | | | | | | | | |
| <1 year | 2,580 | 381,082 | 23,489 | 657,325 | 81.3 (80.6 to 82.1) | 9 | 9,822 | 1,433 | 36,193 | 97.9 (95.9 to 98.9) |
| ≥1 year | 389 | 378,150 | 3,385 | 656,760 | 79.5 (77.1 to 81.5) | 0 | 9,745 | 72 | 36,188 | 100.0 (94.7 to 100.0)[e] |
| **Effectiveness of an Omicron infection in preventing** | | | | | | | | | | |
| **(c) Reinfection with an Omicron virus** | | | | | | **(d) Severe, critical or fatal COVID-19 on reinfection with an Omicron virus** | | | | |
| Any previous infection | 7,476 | 262,515 | 22,170 | 410,785 | 53.6 (52.1 to 55.0) | 0 | 230 | 20 | 630 | 100.0 (79.7 to 100.0)[e] |
| By time since previous infection | | | | | | | | | | |
| Subgroup analysis 1 | | | | | | | | | | |
| 3–<6 months | 647 | 259,038 | 4,924 | 408,074 | 81.3 (79.6 to 82.9) | 0 | 224 | 7 | 630 | 100.0 (30.6 to 100.0)[e] |
| 6–<9 months | 2,611 | 259,888 | 9,689 | 408,881 | 59.8 (57.8 to 61.7) | 0 | 222 | 5 | 630 | 100.0 (−8.4 to 100.0)[e] |
| 9–<1 year | 1,916 | 258,775 | 4,095 | 408,566 | 27.5 (22.7 to 32.0) | 0 | 223 | 3 | 630 | 100.0 (−58.7 to 100.0)[e] |
| ≥1 year | 1,981 | 258,550 | 2,989 | 408,594 | 4.8 (−2.7 to 11.8) | 0 | 225 | 5 | 630 | 100.0 (−8.4 to 100.0)[e] |
| Subgroup analysis 2 | | | | | | | | | | |
| <1 year | 5,318 | 261,806 | 18,950 | 409,948 | 59.5 (58.0 to 60.9) | 0 | 227 | 15 | 630 | 100.0 (72.1 to 100.0)[e] |
| ≥1 year | 1,981 | 258,550 | 2,989 | 408,594 | 4.8 (−2.7 to 11.8) | 0 | 225 | 5 | 630 | 100.0 (−8.4 to 100.0)[e] |

[a]Cases (SARS-CoV-2-positive tests) and controls (SARS-CoV-2-negative tests) were matched exactly one-to-two by sex, 10-year age group, nationality, number of coexisting conditions, number of vaccine doses at time of the SARS-CoV-2 test, calendar week of the SARS-CoV-2 test, method of testing and reason for testing.

[b]Effectiveness of previous infection was estimated using the test-negative, case–control study design[15].

[c]CIs were not adjusted for multiplicity and thus should not be used to infer definitive differences between different groups.

[d]Cases (SARS-CoV-2-positive tests) and controls (SARS-CoV-2-negative tests) were matched exactly one-to-five by sex, 10-year age group, nationality, number of coexisting conditions, number of vaccine doses at time of the SARS-CoV-2 test, calendar week of the SARS-CoV-2 test, method of testing and reason for testing. Severity, criticality and fatality were defined according to the World Health Organization guidelines.

[e]The 95% CI was estimated with the use of McNemar's test because of zero events among exposed cases.

both unvaccinated and vaccinated individuals, as well as effectiveness against severe, critical or fatal COVID-19 on reinfection, and effectiveness over time after the previous infection.

**Validation using a cohort study design**

This study was conducted using the test-negative design[15]. To validate the findings, two national, matched, retrospective cohort studies were also conducted: one for the pre-Omicron era and one for the Omicron era.

The estimates from both the test-negative study design and the cohort study design were consistent across the various outcomes in both the pre-Omicron and Omicron eras (Extended Data Figs. 8 and 9). These outcomes included overall effectiveness against any infection and against infection in both unvaccinated and vaccinated individuals, as well as effectiveness against severe, critical or fatal COVID-19 on reinfection, and effectiveness over time after the previous infection. Whereas the test-negative design yielded lower estimates overall, particularly in the Omicron analysis, both study designs produced similar results for the duration of immune protection, validating the findings of the test-negative design.

**Discussion**

The results show two distinct patterns in the protective effect of natural infection against reinfection in the Omicron era compared to the pre-Omicron era. Before the emergence of Omicron, natural infection offered robust protection against reinfection, with roughly 80% effectiveness and minimal signs of waning over time after the infection. However, during the Omicron era, this protection was strong only for recently infected individuals, rapidly declining over time after the infection and ultimately diminishing within a year. These patterns were consistent regardless of whether any infection or only symptomatic infection was considered as an outcome, and for both vaccinated and unvaccinated populations.

The two distinct patterns observed in the Omicron versus pre-Omicron eras provide population-level results that validate previous experimental molecular evidence[1,2,18–20], and are probably the result of a complex interplay of several interrelated factors, in addition to waning immunity, immune evasion and the accelerated and convergent evolution of Omicron, such as immune imprinting, varying immunogenicity, global

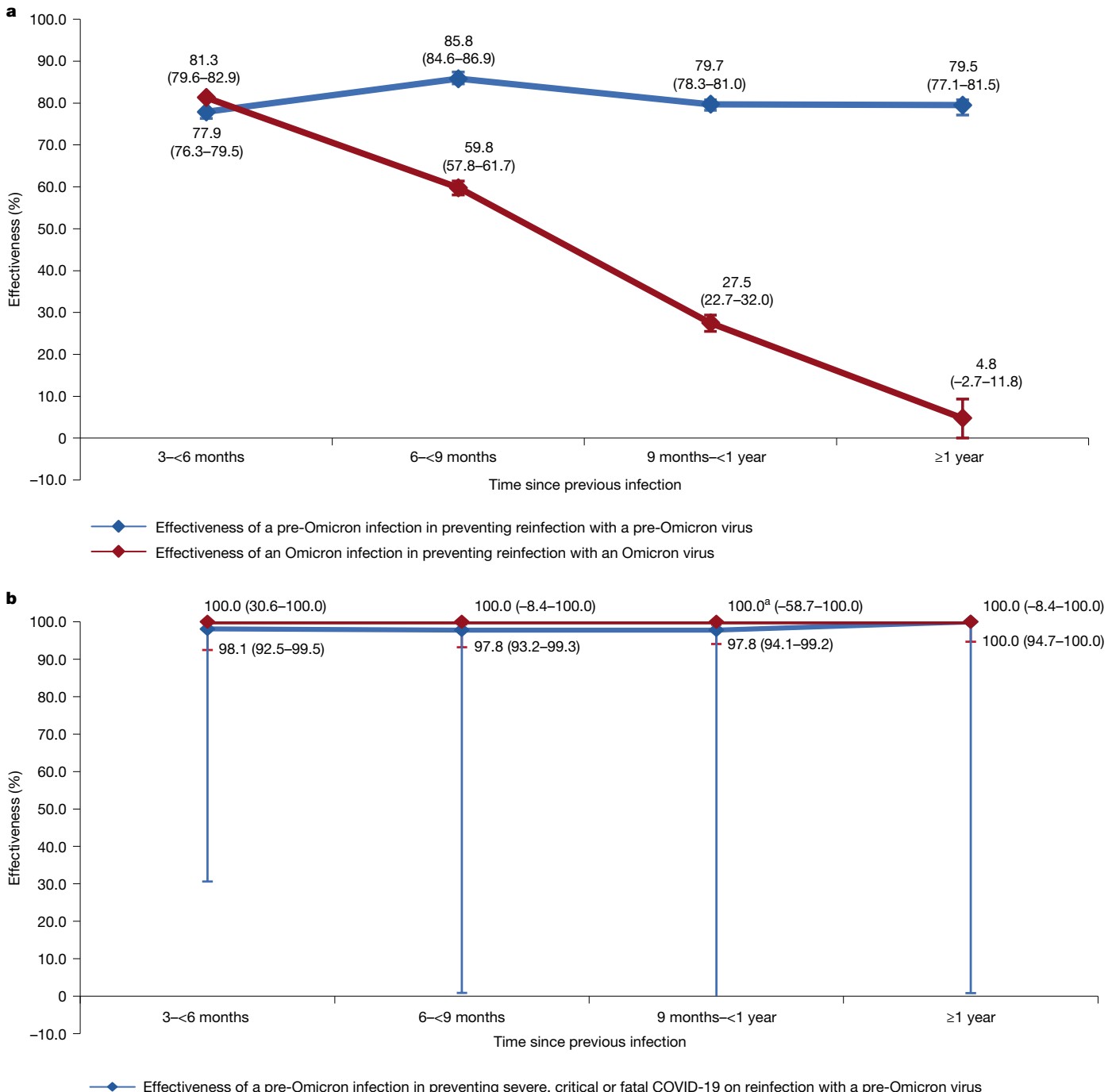

**Fig. 1 | Effectiveness of previous infection against reinfection. a,b,** Effectiveness of infection in preventing reinfection regardless of symptoms (**a**) and in preventing severe, critical or fatal COVID-19 on reinfection (**b**), by time since the previous infection. **a,** Includes 384,231 and 684,257 independent samples for cases and controls, respectively, in the pre-Omicron era analysis, and 269,991 and 432,955 independent samples for each of cases and controls, respectively, in the Omicron era analysis. Panel **b** includes 9,833 and 37,698 independent samples for cases and controls, respectively, in the pre-Omicron era analysis, and 230 and 650 independent samples for each of cases and controls, respectively, in the Omicron era analysis. Data are presented as effectiveness point estimates and corresponding 95% CIs. Error bars indicate the 95% CIs. Measures were not adjusted for multiplicity. In this figure, the 95% CIs are exceedingly small, rendering them barely noticeable for the protection against reinfection in both pre-Omicron and Omicron analyses, as well as for the protection against severe, critical or fatal COVID-19 on reinfection in the pre-Omicron era. This is attributed to the very large sample sizes in these analyses. However, because of the small number of cases of severe forms of COVID-19 in the Omicron era, the 95% CIs are very wide for the protection against severe, critical or fatal COVID-19 on reinfection in the Omicron era. [a]The negative lower bound for the CI was truncated because the CI was too wide.

population immunity faced by the strains and population characteristics associated with infections at different stages of the pandemic.

Whereas these factors are interconnected and challenging to disentangle, the observed differences in protection against reinfection may stem from distinct evolutionary pressures acting on SARS-CoV-2 during the pre-Omicron and Omicron eras. In the pre-Omicron era, with a large proportion of individuals remaining immune naive because of non-pharmaceutical interventions and delayed scale-up of vaccination,

**Table 2 | Effectiveness of previous infection against symptomatic reinfection**

| Effectiveness | Cases[a] | | Controls[a] | | Effectiveness[b] (%) (95% CI)[c] |
|---|---|---|---|---|---|
| | Previous infection (n) | No previous infection (n) | Previous infection (n) | No previous infection (n) | |
| **(a) Effectiveness of a pre-Omicron infection against symptomatic[d] reinfection with a pre-Omicron virus** | | | | | |
| Any previous infection | 626 | 139,378 | 7,277 | 212,793 | 86.8 (85.7 to 87.9) |
| By time since previous infection | | | | | |
| Subgroup analysis 1 | | | | | |
| 3–<6 months | 177 | 137,936 | 2,027 | 212,561 | 85.2 (82.8 to 87.4) |
| 6–<9 months | 164 | 138,182 | 2,375 | 212,594 | 89.4 (87.5 to 91.0) |
| 9 months–<1 year | 236 | 138,453 | 2,116 | 212,662 | 84.6 (82.4 to 86.6) |
| ≥1 year | 44 | 137,707 | 753 | 212,536 | 90.0 (86.5 to 92.7) |
| Subgroup analysis 2 | | | | | |
| <1 year | 581 | 139,316 | 6,523 | 212,781 | 86.5 (85.3 to 87.6) |
| ≥1 year | 44 | 137,707 | 753 | 212,536 | 90.0 (86.5 to 92.7) |
| **(b) Effectiveness of an Omicron infection against symptomatic[d] reinfection with an Omicron virus** | | | | | |
| Any previous infection | 3,211 | 49,371 | 7,629 | 62,670 | 45.4 (42.5 to 48.2) |
| By time since previous infection | | | | | |
| Subgroup analysis 1 | | | | | |
| 3–<6 months | 119 | 47,753 | 1,125 | 61,367 | 86.3 (83.2 to 88.9) |
| 6–<9 months | 606 | 48,046 | 2,236 | 61,610 | 64.0 (60.1 to 67.5) |
| 9 months–<1 year | 834 | 47,835 | 1,781 | 61,696 | 29.8 (22.6 to 36.4) |
| ≥1 year | 1,481 | 47,952 | 2,222 | 61,971 | 4.8 (−3.8 to 12.7) |
| Subgroup analysis 2 | | | | | |
| <1 year | 1,609 | 48,837 | 5,244 | 62,054 | 58.9 (56.1 to 61.5) |
| ≥1 year | 1,481 | 47,952 | 2,222 | 61,971 | 4.8 (−3.8 to 12.7) |

[a]Cases (SARS-CoV-2-positive tests) and controls (SARS-CoV-2-negative tests) were matched exactly one-to-two by sex, 10-year age group, nationality, number of coexisting conditions, number of vaccine doses at time of the SARS-CoV-2 test, calendar week of the SARS-CoV-2 test, method of testing and reason for testing.
[b]Effectiveness of previous infection in preventing reinfection was estimated using the test-negative, case–control study design[15].
[c]CIs were not adjusted for multiplicity and thus should not be used to infer definitive differences between different groups.
[d]A symptomatic infection was defined as a SARS-CoV-2 PCR or rapid antigen test conducted because of clinical suspicion due to the presence of symptoms compatible with a respiratory tract infection.

intrinsic transmissibility may have been the primary driver of viral adaptation. This was evidenced by the emergence of more transmissible variants such as Alpha[4,22,23] and Delta[24,25]. Conversely, following the very large and widespread Omicron wave in early 2022 (Extended Data Fig. 3)[26], most individuals possessed some level of immunity, either from infection or vaccination. This may have shifted the dominant evolutionary pressure towards immune escape through not only antigenic

drift, but also recombination and convergent evolution as the adaptive mechanisms for the virus[2,18,27,28].

This shift in evolutionary pressure can explain the observed rapid decline in natural immunity protection against reinfection in the Omicron era and suggests an accelerated evolution of SARS-CoV-2 towards enhanced immune evasion in the current stage of the pandemic. The extent to which these findings describe an ecological survival strategy for novel respiratory viruses capable of rapid mutation, such as RNA viruses, as they transition from emergence to endemicity remains unknown, as does the relevance of these findings to other circulating respiratory viruses.

Immune imprinting, the memory recall behaviour of the immune system, in which the outcome is influenced by the antigenic distance between the ancestral strain and the variant[27,29–32], may also help to explain these findings. This phenomenon, in which the specific sequence of immunological events can either enhance or compromise future immune responses to variant infections, may shape the diversity of polyclonal neutralizing antibodies elicited by Omicron breakthrough infections, potentially contributing to the observed patterns[27,29–33].

An important finding is the robust and durable protection against severe COVID-19 on reinfection in both pre-Omicron and Omicron eras, with no observed waning in this protection. This distinct pattern suggests the involvement of different immune system components in protecting against non-severe versus severe reinfection. Whereas humoral immunity, mediated by neutralizing antibodies that block viral entry[34,35], is the primary driver of protection against non-severe reinfection, it faces intense pressure from viral evolution towards immune escape. Conversely, protection against severe reinfection seems strongly influenced by cellular immunity through memory T cells[36–38], which appears to be largely conserved[37,38], perhaps indicating limited pressure on cellular immunity from the evolutionary forces driving immune escape.

Both vaccinated and unvaccinated populations showed similar contrasting patterns of protection between the pre-Omicron and Omicron eras (Extended Data Fig. 4). However, small quantitative differences in effectiveness were observed between these groups. These differences might be attributed to variations in the distribution of time intervals between the previous infection and the study test, a consequence of the changing dynamics of infection waves and vaccination scale-up over time. Previous studies suggest that immune imprinting effects might also have a role in explaining these differences[11,12,30], but further investigation is needed to clarify these patterns.

Our findings at this stage of the pandemic corroborate earlier observations, demonstrating the consistently reduced severity of reinfections compared to primary infections[9,14,39], even when both involve the Omicron variant. The results also confirm the strong protective effect of pre-Omicron immunity against pre-Omicron virus[4,7–10,40,41]. Furthermore, they highlight an association between viral immune evasion and accelerated waning of immune protection. This accelerated waning was previously observed in the rapid decline of pre-Omicron immunity, induced by both vaccination or natural infection, against the Omicron variant on its emergence[14,42–45]. Collectively, these findings suggest that immune evasion and subsequent rapid waning of immunity hinder the development of long-term population immunity to SARS-CoV-2, potentially leading to periodic waves of infection[46], similar to those observed for common-cold coronaviruses[47,48] and influenza[49,50].

This study has limitations. The study is based on documented SARS-CoV-2 infections, but many infections may not have been documented, especially since the reduction in testing starting from 1 November 2022. However, this under-documentation of infection, a source of misclassification bias of previous infection status, may not have appreciably affected our findings based on our earlier analysis of the test-negative design methodology[15]. It should not have affected the estimated duration of immune protection, as demonstrated in the extra mathematical modelling analyses conducted in this study (Extended Data Fig. 5).

**Table 3 | Effectiveness of previous infection against reinfection by vaccination status**

| Effectiveness | Cases[a] | | Controls[a] | | Effectiveness[b] in (%) (95% CI)[c] |
|---|---|---|---|---|---|
| | Previous infection (n) | No previous infection (n) | Previous infection (n) | No previous infection (n) | |
| **(a) Effectiveness of a pre-Omicron infection against reinfection with a pre-Omicron virus** | | | | | |
| **Including only unvaccinated individuals** | | | | | |
| Any previous infection | 2,247 | 351,238 | 20,768 | 607,068 | 81.5 (80.6 to 82.3) |
| By time since previous infection | | | | | |
| Subgroup analysis 1 | | | | | |
| 3–<6 months | 665 | 349,165 | 5,938 | 606,661 | 79.9 (78.2 to 81.5) |
| 6–<9 months | 501 | 349,444 | 6,679 | 606,712 | 87.1 (85.8 to 88.2) |
| 9 months–<1 year | 846 | 349,835 | 6,399 | 606,804 | 78.4 (76.8 to 80.0) |
| ≥1 year | 219 | 348,655 | 1,736 | 606,573 | 76.8 (73.3 to 79.9) |
| Subgroup analysis 2 | | | | | |
| <1 year | 2,025 | 351,163 | 19,029 | 607,049 | 81.9 (81.0 to 82.7) |
| ≥1 year | 219 | 348,655 | 1,736 | 606,573 | 76.8 (73.3 to 79.9) |
| **Including only vaccinated individuals** | | | | | |
| Any previous infection | 726 | 30,020 | 6,112 | 50,309 | 79.9 (78.2 to 81.4) |
| By time since previous infection | | | | | |
| Subgroup analysis 1 | | | | | |
| 3–<6 months | 205 | 29,426 | 1,080 | 50,163 | 67.2 (61.8 to 71.8) |
| 6–<9 months | 185 | 29,536 | 1,651 | 50,181 | 80.6 (77.4 to 83.4) |
| 9 months–<1 year | 161 | 29,570 | 1,725 | 50,168 | 84.6 (81.8 to 86.9) |
| ≥1 year | 170 | 29,495 | 1,649 | 50,187 | 82.2 (79.0 to 84.8) |
| Subgroup analysis 2 | | | | | |
| <1 year | 555 | 29,919 | 4,460 | 50,276 | 79.0 (77.0 to 80.8) |
| ≥1 year | 170 | 29,495 | 1,649 | 50,187 | 82.2 (79.0 to 84.8) |
| **(b) Effectiveness of an Omicron infection against reinfection with an Omicron virus** | | | | | |
| **Including only unvaccinated individuals** | | | | | |
| Any previous infection | 1,686 | 92,144 | 4,899 | 152,798 | 47.1 (43.7 to 50.4) |
| By time since previous infection | | | | | |
| Subgroup analysis 1 | | | | | |
| 3–<6 months | 266 | 91,534 | 1,290 | 152,302 | 68.0 (63.1 to 72.2) |
| 6–<9 months | 778 | 91,662 | 2,328 | 152,492 | 46.0 (40.9 to 50.7) |
| 9 months–<1 year | 331 | 91,418 | 708 | 152,298 | 25.5 (14.0 to 35.5) |
| ≥1 year | 265 | 91,376 | 513 | 152,277 | 25.9 (11.7 to 37.8) |
| Subgroup analysis 2 | | | | | |
| <1 year | 1,404 | 92,030 | 4,371 | 152,704 | 49.4 (45.9 to 52.7) |
| ≥1 year | 265 | 91,376 | 513 | 152,277 | 25.9 (11.7 to 37.8) |
| **Including only vaccinated individuals** | | | | | |
| Any previous infection | 5,790 | 170,371 | 17,271 | 257,987 | 55.5 (53.9 to 57.1) |
| By time since previous infection | | | | | |
| Subgroup analysis 1 | | | | | |
| 3–<6 months | 381 | 167,504 | 3,634 | 255,772 | 85.8 (84.0 to 87.3) |
| 6–<9 months | 1,833 | 168,226 | 7,361 | 256,389 | 64.0 (61.9 to 66.0) |
| 9 months–<1 year | 1,585 | 167,357 | 3,387 | 256,268 | 28.0 (22.7 to 32.9) |
| ≥1 year | 1,716 | 167,174 | 2,476 | 256,317 | −1.2 (−9.3 to 7.0) |
| Subgroup analysis 2 | | | | | |
| <1 year | 3,914 | 169,776 | 14,579 | 257,244 | 62.5 (60.9 to 64.0) |
| ≥1 year | 1,716 | 167,174 | 2,476 | 256,317 | −1.2 (−9.3 to 7.0) |

Continued

[a]Cases (SARS-CoV-2-positive tests) and controls (SARS-CoV-2-negative tests) were matched exactly one-to-two by sex, 10-year age group, nationality, number of coexisting conditions, number of vaccine doses at time of the SARS-CoV-2 test, calendar week of the SARS-CoV-2 test, method of testing and reason for testing.
[b]Effectiveness of previous infection in preventing reinfection was estimated using the test-negative, case–control study design[15].
[c]CIs were not adjusted for multiplicity and thus should not be used to infer definitive differences between different groups.

Depletion of the primary-infection cohort within the population owing to COVID-19 mortality at the time of primary infection can skew this cohort towards healthier individuals[51]. This skew could potentially lead to an overestimation of the effectiveness of infection in preventing severe, critical or fatal COVID-19 on reinfection. Nevertheless, given the low COVID-19 mortality rate in Qatar's predominantly young population (less than 0.1% of primary infections)[26,52,53], a survival effect is not likely to appreciably alter the observed high effectiveness of previous infection in mitigating the severity of reinfection.

In the sensitivity analysis using a 40-day time window to define reinfection, as opposed to the conventional 90-day time window[21] (Extended Data Table 2), estimates for both overall protection and protection lasting less than 1 year showed a slight bias towards lower values. This bias stemmed from the inclusion of SARS-CoV-2 tests conducted between 40 and 90 days post-initial infection. Tests within this timeframe captured sporadic positive PCR results that did not signify genuine reinfections but rather reflected instances in which the initial infection persisted beyond 40 days or in which PCR positivity persisted because of residual non-viable viral fragments[54]. This source of bias, however, should not affect other estimates in this study as it is very rare for such false reinfections to occur after 90 days post-initial infection[21].

A consequence of the test-negative design is that effectiveness estimates depend on the timing of the tests, specifically the interval between the previous infection and the SARS-CoV-2 test used in the study[15]. This can lead to seemingly contradictory results that are not necessarily indicative of true discrepancies. For instance, in the Omicron era analysis, the effectiveness against any reinfection (53.6%) was higher than that against symptomatic reinfection (45.4%). This might seem counterintuitive, as protection against symptomatic illness is typically expected to be higher[55]. However, this apparent discrepancy can be explained by the differential timing between the previous infection and the study test.

The median time between the previous infection and the study test was 245 days for the analysis of any reinfection, whereas it was longer at 301 days for the analysis of symptomatic reinfection. This longer interval in the latter analysis resulted in greater waning of immune protection, leading to the observed difference in effectiveness estimates.

With the relatively young population of Qatar[17], our findings may not be generalizable to other countries in which older citizens constitute a large proportion of the population. Whereas robust matching was implemented, the availability of data prevented matching on other factors such as geography or occupation. However, being essentially a city state, infection incidence in Qatar was broadly distributed across neighbourhoods. Nationality, age and sex provide a powerful proxy for socioeconomic status in this country[17], and thus matching by these factors may have also, at least partially, controlled for other factors such as occupation. This matching approach has been previously investigated in studies of different epidemiologic designs, and using control groups to test for null effects[56–59]. These studies have supported that this matching prescription effectively controls for differences in infection exposure[56–59].

However, bias in real-world data can arise unexpectedly or from unknown sources, such as subtle differences in test-seeking behaviour, changes in testing patterns because of policy shifts, variations in test accessibility or differences in the tendency to get tested between individuals who have recovered from a previous infection and those who have not been infected or whose previous infection was undocumented. Nevertheless, the further analyses investigating the impact of different sources of bias and the replication of the entire study using a different study design—a cohort study—confirmed and validated the findings.

This study has strengths. First, it was conducted on a national scale, encompassing a diverse population based on national backgrounds, and used extensive and validated databases established through numerous SARS-CoV-2 infection studies. Second, controls were selected from the entire national population, and exact matching was used to ensure rigorous pairing of cases and controls. Third, the study design controlled for vaccination status, another strength of the test-negative design, allowing for differentiation between the effects of previous infection and vaccination. Effectiveness estimates were also obtained for both unvaccinated and vaccinated populations, and the results of these analyses were consistent with each other and with the main analysis for the entire population. Fourth, using distinct reinfection window definitions of 90 versus 40 days resulted in considerably different analysis datasets. Yet, the effectiveness estimates remained consistent across these varying criteria. Finally, the findings were corroborated through further analyses and by a complete replication using a distinct epidemiological study design.

In conclusion, a stark contrast is found in the protective effect of natural infection against SARS-CoV-2 reinfection between the pre-Omicron and Omicron eras. Before Omicron emergence, natural infection provided robust and enduring protection against reinfection. However, during the Omicron era, this protection was strong only among recently infected individuals, rapidly declining and diminishing within 1 year. These contrasting patterns may suggest differing evolutionary pressures acting on the virus. In the pre-Omicron era, intrinsic transmissibility drove viral adaptation, whereas the widespread immunity acquired by the end of the first Omicron wave shifted the evolutionary pressure towards immune escape. This highlights the dynamic interplay between viral evolution and host immunity, necessitating continued monitoring of the virus and its evolution, as well as periodic updates of SARS-CoV-2 vaccines to restore immunity and counter continuing viral immune evasion.

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

## Methods

### Oversight

The institutional review boards at Hamad Medical Corporation and Weill Cornell Medicine–Qatar approved this retrospective study with a waiver of informed consent. The study was reported according to the Strengthening the Reporting of Observational Studies in Epidemiology (STROBE) guidelines (Extended Data Table 3).

### Study population and data sources

This study was conducted on the population of Qatar before and after the introduction of the Omicron variant on 19 December 2021 (ref. 16). The first analysis assessed the effectiveness of pre-Omicron infection in preventing reinfection with a pre-Omicron virus between 5 February 2020 (the onset of the COVID-19 pandemic in Qatar[17]) and 18 December 2021. The second analysis assessed the effectiveness of Omicron infection in preventing reinfection with an Omicron virus between 19 December 2021 and 12 February 2024 (marking the end of the study).

The data encompassed the national, federated databases for COVID-19 laboratory testing, vaccination, hospitalization and death, retrieved from the integrated, nationwide, digital-health information platform (Supplementary Methods section 2). These databases have captured SARS-CoV-2-related data with no missing information since the onset of the pandemic, including all PCR tests regardless of location or facility, and, from 5 January 2022, all medically supervised rapid antigen tests (Supplementary Methods section 3). SARS-CoV-2 testing was extensive in Qatar until 31 October 2022, with nearly 5% of the population being tested every week, primarily for routine purposes such as screening or meeting travel-related requirements[56,60]. Subsequently, testing rates decreased, with less than 1% of the population being tested per week[61]. Most infections during the pandemic were diagnosed through routine testing rather than symptomatic presentation[56,60].

Qatar launched its COVID-19 vaccination programme in December 2020, using messenger RNA vaccines and prioritizing individuals on the basis of coexisting conditions and age criteria[56,59]. COVID-19 vaccination was provided free of charge, regardless of citizenship or residency status, and was nationally tracked[56,59]. Demographic information, including sex, age and nationality, was extracted from the records of the national health registry. Qatar shows demographic diversity, with 89% of its residents being expatriates from more than 150 countries[17]. Detailed descriptions of Qatar's population and national databases have been previously reported[17,32,51,53,56,60,62].

### Study design

The effectiveness of natural infection against reinfection was estimated using the test-negative, case-control study design, which compares the odds of previous infection among SARS-CoV-2-positive tests (cases) to that among SARS-CoV-2-negative tests (controls)[11,12,15,16,63,64]. The assessment was conducted both overall and by time since previous infection, using 3-month intervals.

Cases and controls were determined on the basis of the results of SARS-CoV-2 tests conducted during each analysis period. Cases were defined as SARS-CoV-2-positive tests, whereas controls were defined as SARS-CoV-2-negative tests. SARS-CoV-2 reinfection is conventionally defined as a documented infection more than or equal to 90 days after a previous infection, to avoid misclassifying prolonged test positivity as reinfection with shorter time intervals[16,21,65,66]. Consequently, cases or controls preceded by SARS-CoV-2-positive tests within 90 days were excluded. Previous infection was defined as a SARS-CoV-2-positive test more than or equal to 90 days before the study test.

To comply with the non-differential healthcare-seeking behaviour assumption inherent to the test-negative study design[15,63,64], only tests with a documented reason for testing were included in the analysis. In the Omicron era analysis, cases or controls preceded by a pre-Omicron infection were excluded from the analysis, as the research question pertained to the effectiveness of Omicron immunity, rather than pre-Omicron immunity, against Omicron reinfection. The protection provided by pre-Omicron immunity against Omicron reinfection has been previously investigated[14,16,60].

In estimating effectiveness in preventing reinfection, cases and controls were matched exactly one-to-two by sex, 10-year age group, nationality, number of coexisting conditions (ranging from zero to more than or equal to six; Supplementary Methods section 1), number of vaccine doses (ranging from zero to more than or equal to four), calendar week of the SARS-CoV-2 test, method of testing (PCR or rapid antigen) and reason for testing. This matching strategy aimed to balance observed confounders that could potentially influence the risk of infection across the exposure groups[17,67–70]. The selection of matching factors was guided by findings from earlier studies on Qatar's population[11,56–59,71] and the need to comply with the non-differential healthcare-seeking behaviour assumption inherent to the test-negative design[15,63,64]. This requirement was met by matching by the calendar week of the SARS-CoV-2 test, method of testing and reason for testing. In estimating effectiveness in preventing severe[72], critical[72] or fatal[73] COVID-19 on reinfection, a one-to-five matching ratio was applied to enhance statistical precision.

Classification of severe[72], critical[72] and fatal[73] COVID-19 followed the World Health Organization guidelines (Supplementary Methods section 4). The assessments were made by trained medical personnel independent of study investigators and using individual chart reviews. As part of the national protocol, each individual who had a SARS-CoV-2-positive test and concurrent COVID-19 hospital admission was subject to an infection severity assessment every 3 days until discharge or death, irrespective of hospital length of stay[26]. Individuals who progressed to severe, critical or fatal COVID-19 between the SARS-CoV-2-positive test and the end of this study were classified on the basis of their worst outcome, starting with death, followed by critical disease and then severe disease.

The variant status of infections was determined on the basis of the dominant variant at the time of infection diagnosis. Infections were categorized as pre-Omicron or Omicron. The duration of dominance for every variant throughout the pandemic was determined using Qatar's variant genomic surveillance[74–76] (Extended Data Fig. 3), which includes viral genome sequencing[74] and multiplex real-time quantitative PCR with reverse transcription (RT–qPCR) variant screening[75] of weekly collected random positive clinical samples (Supplementary Methods section 3). Once the first massive Omicron wave started, virtually all infections were due to Omicron, as it displaced all pre-Omicron variants[16,60,62].

### Statistical analysis

All records of SARS-CoV-2 testing were examined for the selection of cases and controls, but only matched samples were analysed. Cases and controls were described using frequency distributions and measures of central tendency and compared using standardized mean differences. A standardized mean difference of less than or equal to 0.1 indicated adequate matching[77].

Odds ratios (ORs), comparing odds of previous infection among cases versus controls, and associated 95% CIs were derived using conditional logistic regression. Analyses stratified by time since previous infection considered the date for the most recent documented infection. CIs were not adjusted for multiplicity and interactions were not investigated. The reference group for all estimates comprised individuals with no documented previous infection.

Effectiveness measures and associated 95% CIs were calculated as 1-OR of previous infection among cases versus controls if the OR was less than one, and as 1/OR-1 if the OR was more than or equal to one (refs. 15,32,63,78). This approach ensured a symmetric scale for both negative and positive effectiveness, spanning from −100 to 100%, resulting in a clear and meaningful interpretation of effectiveness, regardless of the value being positive or negative.

In addition to estimating effectiveness of previous infection in preventing reinfection, regardless of symptoms, effectiveness was also assessed specifically against symptomatic reinfection. This was accomplished by restricting the analysis to tests performed owing to clinical suspicion, indicating the presence of symptoms consistent with a respiratory tract infection.

Subgroup analyses were performed, considering only unvaccinated and vaccinated individuals, respectively. A sensitivity analysis was undertaken by redefining SARS-CoV-2 reinfection as a documented infection occurring more than or equal to 40 days after a previous infection, instead of the conventional more than or equal to 90 days. This adjustment was informed by a recent analysis suggesting the adequacy of a 40-day time window to define reinfection[21]. Statistical analyses were conducted in STATA/SE version v.18.0 (Stata).

### Further validation analyses

**Previous infection misclassification.** Under-ascertainment of infection introduces misclassification of previous infection status into the test-negative design used in this study, potentially biasing the estimates[15]. To assess the impact of under-ascertainment on estimates of waning immune protection, mathematical modelling simulations were conducted by extending our previous work on the test-negative design methodology[15]. These simulations evaluated the impact of an infection ascertainment rate of only 10%, indicating that 90% of SARS-CoV-2 infections are undocumented and would thus be misclassified. The model incorporated a gradual (linear) waning of the protective effect of infection against reinfection. Analyses were performed for both pre-Omicron and Omicron waning patterns, with immune protection durations set at 3 years for the pre-Omicron era[14] and 1 year for the Omicron era. The results were reported at the 2-year mark from the onset of both the pre-Omicron and Omicron pandemic phases. A detailed description of the model, its methods and previous analyses can be found elsewhere[15].

**Coexisting conditions misclassification.** Coexisting conditions were identified by analysing electronic health record encounters for each individual within the national public healthcare system's database (Supplementary Methods section 1). However, this approach may not capture all conditions, as some may be undiagnosed or diagnosed at private facilities with unavailable records. To assess the impact of this potential bias on the estimated effectiveness of infection against reinfection, a sensitivity analysis was conducted in which matching by the number of coexisting conditions was entirely removed, simulating a scenario of complete under-ascertainment of these conditions.

**Validation using a cohort study design.** This study used the test-negative design[15]. To validate the findings, two extra national, matched, retrospective cohort studies—one for the pre-Omicron era and another for the Omicron era—were conducted. Each study compared the incidence of infection and severe forms of COVID-19 in two national cohorts: individuals with documented primary SARS-CoV-2 infection (primary-infection cohort) and uninfected individuals (uninfected cohort).

The first study estimated the effectiveness of a pre-Omicron primary infection in preventing reinfection with a pre-Omicron virus, and the second study estimated the effectiveness of an Omicron primary infection in preventing reinfection with an Omicron virus. Both studies were conducted over the same study durations as in the main analysis using the test-negative design. Incidence of infection was defined as any PCR-positive or rapid antigen-positive test after the start of follow-up, irrespective of symptomatic presentation.

Cohorts were matched exactly one-to-one by sex, 10-year age group, nationality, number of coexisting conditions, number of vaccine doses at the start of the follow-up, calendar week of the SARS-CoV-2-positive test for the primary-infection cohort and SARS-CoV-2-negative test for the uninfected cohort, method of testing (PCR or rapid antigen) and reason for testing. In both studies, individuals in the matched primary-infection cohort may have contributed follow-up time in the uninfected cohort before their primary infection and subsequently contributed follow-up time as part of the primary-infection cohort after contracting the infection.

Follow-up for each matched pair started 90 days after the primary infection for the individual in the primary-infection cohort. To ensure exchangeability, both members of each matched pair were censored at the earliest occurrence of receiving an extra vaccine dose. Accordingly, individuals were followed until the first of any of the following events: a documented SARS-CoV-2 infection, a new vaccine dose for the individual in either the primary-infection cohort or the uninfected cohort (with matched-pair censoring), death or the administrative end of follow-up, which was set at the end of the study or 15 months after the primary infection, whichever came first.

The overall adjusted hazard ratio (aHR), comparing the incidence of SARS-CoV-2 infection (or severe forms of COVID-19) between the cohorts, and the corresponding 95% CI, were calculated using Cox regression models with adjustment for the matching factors and testing rate in the cohorts, using the Stata v.18.0 stcox command. This adjustment was implemented to ensure precise and unbiased estimation of the standard variance[79].

The overall aHR provides a weighted average of the time-varying hazard ratio[80]. To explore differences in the risk of infection (or severe forms of COVID-19) over time, the aHR was also estimated by 3-month intervals from the start of follow-up using separate Cox regressions, with 'failure' restricted to specific time intervals.

Effectiveness of infection against reinfection and against severe, critical or fatal COVID-19, along with the associated 95% CIs, were derived from the aHR as 1-aHR if the aHR was less than one and as 1/aHR-1 if the aHR was more than or equal to one (refs. 32,78). This approach ensured a symmetric scale for both negative and positive effectiveness, spanning from −100 to 100%, resulting in a meaningful interpretation of effectiveness, regardless of the value being positive or negative.

Statistical analyses were performed using Stata/SE v.18.0 (Stata). Further details on this type of cohort study design can be found in our previous publications, which used also the same national databases to estimate the effectiveness of infection against reinfection or the effectiveness of vaccination against infection[4,8,10,14,15,30,32,33,45,59,61,62,71,81–83].

### Reporting summary

Further information on research design is available in the Nature Portfolio Reporting Summary linked to this article.

## Data availability

The National Coronavirus Disease 2019 (COVID-19) dataset used in this study is a property of the Qatar Ministry of Public Health that was provided to the researchers through a restricted-access agreement that prevents sharing the dataset with a third party or publicly. This dataset encompasses the National COVID-19 Testing Database, the National COVID-19 Vaccination Database, the National COVID-19 Severity Database, and the National Mortality Database. These data are available under restricted access for preservation of confidentiality of patient data. Access can be obtained through a direct application for data access to Her Excellency the Minister of Public Health (https://www.moph.gov.qa/english/OurServices/eservices/Pages/Governmental-HealthCommunication-Center.aspx). Data were available to authors through .csv files in which information has been downloaded from the CERNER database system (no links/accession codes were available to authors). The raw data are protected and are not available owing to data privacy laws. Aggregate data are available within the paper and its Supplementary information.

## Code availability

Standard epidemiological analyses were conducted using standard commands in STATA/SE v.18.0. These commands have been published at: https://github.com/IDEGWCMQ/TestNegCode/blob/main/Test-NegCode.txt.

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

**Acknowledgements** We acknowledge the many dedicated individuals at Hamad Medical Corporation, the Ministry of Public Health, the Primary Health Care Corporation, Qatar Biobank, Sidra Medicine and Weill Cornell Medicine–Qatar for their efforts and contributions to make this study possible. We are also grateful for institutional salary support from the Biomedical Research Programme and the Biostatistics, Epidemiology and Biomathematics Research Core, both at Weill Cornell Medicine–Qatar, as well as for institutional salary support provided by the Ministry of Public Health, Hamad Medical Corporation and Sidra Medicine. We are also grateful for the Qatar Genome Programme and Qatar University Biomedical Research Center for institutional support for the reagents needed for the viral genome sequencing. The funders of the study had no role in study design, data collection, data analysis, data interpretation or writing of the article. Statements made herein are solely the responsibility of the authors.

**Author contributions** H.C. co-designed the study, performed the statistical analyses and co-wrote the first draft of the article. L.J.A.-R. conceived and co-designed the study, led the statistical analyses and co-wrote the first draft of the article. H.C. and L.J.A.-R. accessed and verified all the data. P.C. designed mass PCR testing to allow routine capture of variants and conducted viral genome sequencing. P.T. and M.R.H. designed and conducted multiplex, RT–qPCR variant screening and viral genome sequencing. H.M.Y. and A.A.A.T. conducted viral genome sequencing. H.C., H.H.A., P.C., P.T., M.R.H., H.M.Y., A.A.A.T., Z.A.-K., E.A.-K., A.J., A.H.K., A.N.L., R.M.S., H.F.A.-R., G.K.N., M.G.A.-K., A.A.B., H.E.A.-R., M.H.A.-T., A.A.-K., R.B. and L.J.A.-R. contributed to data collection and acquisition, database development, discussion and interpretation of the results, writing of the article and read and approved the final manuscript.

**Competing interests** A.A.B. has received institutional grant funding from Gilead Sciences unrelated to the work presented in this paper. The other authors declare no competing interests.

**Additional information**
**Correspondence and requests for materials** should be addressed to Hiam Chemaitelly or Laith J. Abu-Raddad.

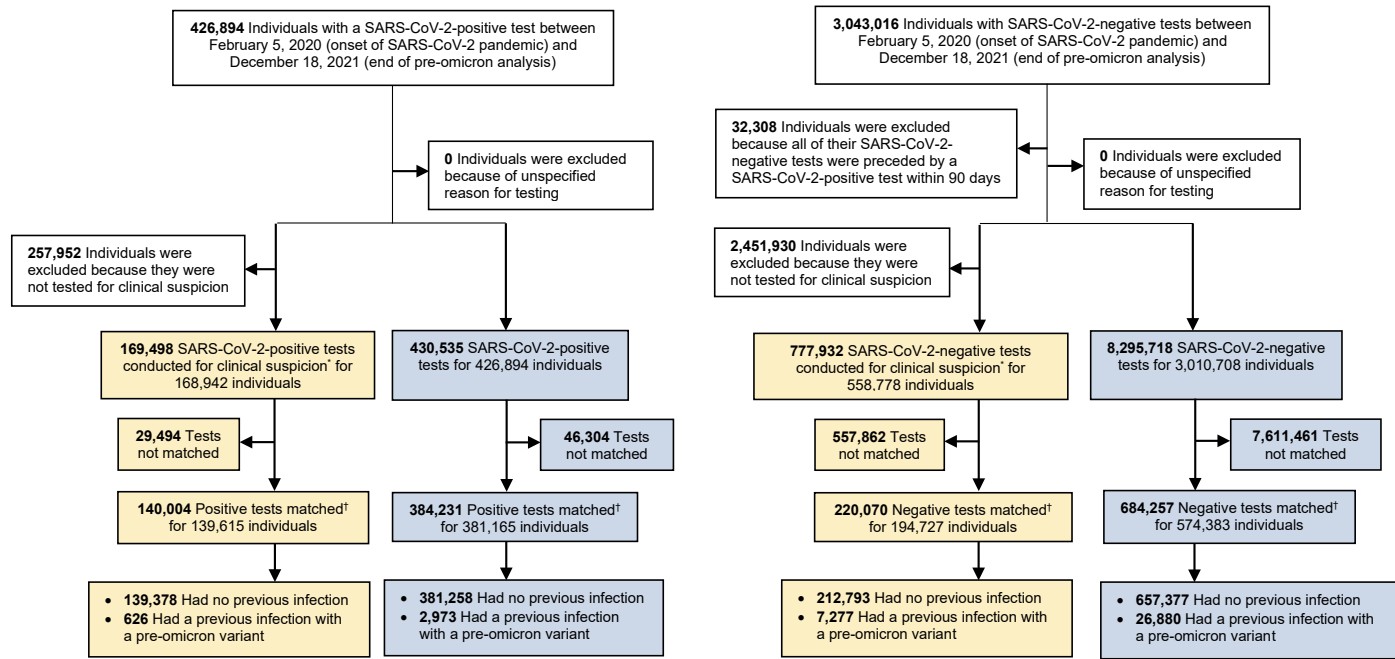

*A symptomatic infection was defined as a severe acute respiratory syndrome coronavirus 2 (SARS-CoV-2) polymerase chain reaction (PCR) or rapid antigen (RA) test conducted because of clinical suspicion due to presence of symptoms compatible with a respiratory tract infection.

†SARS-CoV-2 positives tests were matched one-to-two to SARS-CoV-2 negative tests by sex, 10-year age group, nationality, number of coexisting conditions, number of vaccine doses at time of the SARS-CoV-2 test, calendar week of SARS-CoV-2 test, method of testing (PCR or RA), and reason for testing.

**Extended Data Fig. 1 | Study population selection process in the pre-omicron analysis.** Flowchart describing the population selection process for investigating the effectiveness of a pre-omicron infection in preventing reinfection with a pre-omicron virus.

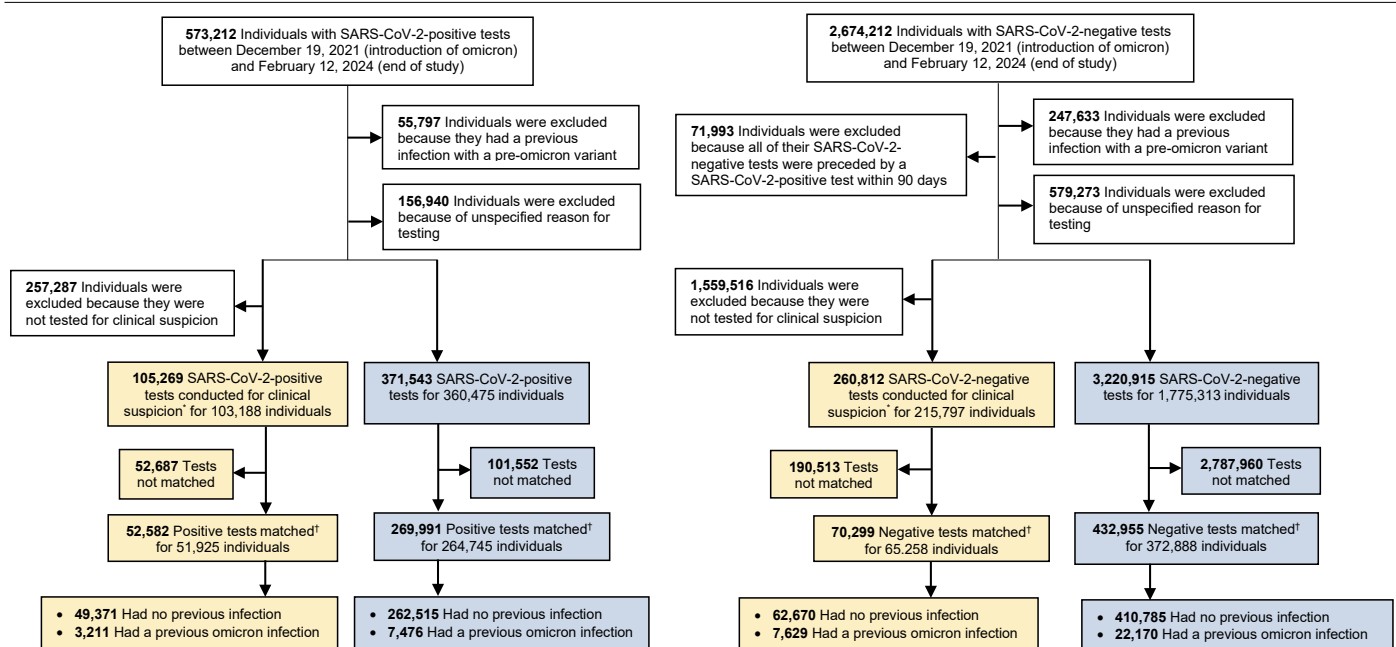

\*A symptomatic infection was defined as a severe acute respiratory syndrome coronavirus 2 (SARS-CoV-2) polymerase chain reaction (PCR) or rapid antigen (RA) test conducted because of clinical suspicion due to presence of symptoms compatible with a respiratory tract infection.

†SARS-CoV-2 positives tests were matched one-to-two to SARS-CoV-2 negative tests by sex, 10-year age group, nationality, number of coexisting conditions, number of vaccine doses at time of the SARS-CoV-2 test, calendar week of SARS-CoV-2 test, method of testing (PCR or RA), and reason for testing.

**Extended Data Fig. 2 | Study population selection process in the omicron analysis.** Flowchart describing the population selection process for investigating the effectiveness of an omicron infection in preventing reinfection with an omicron virus.

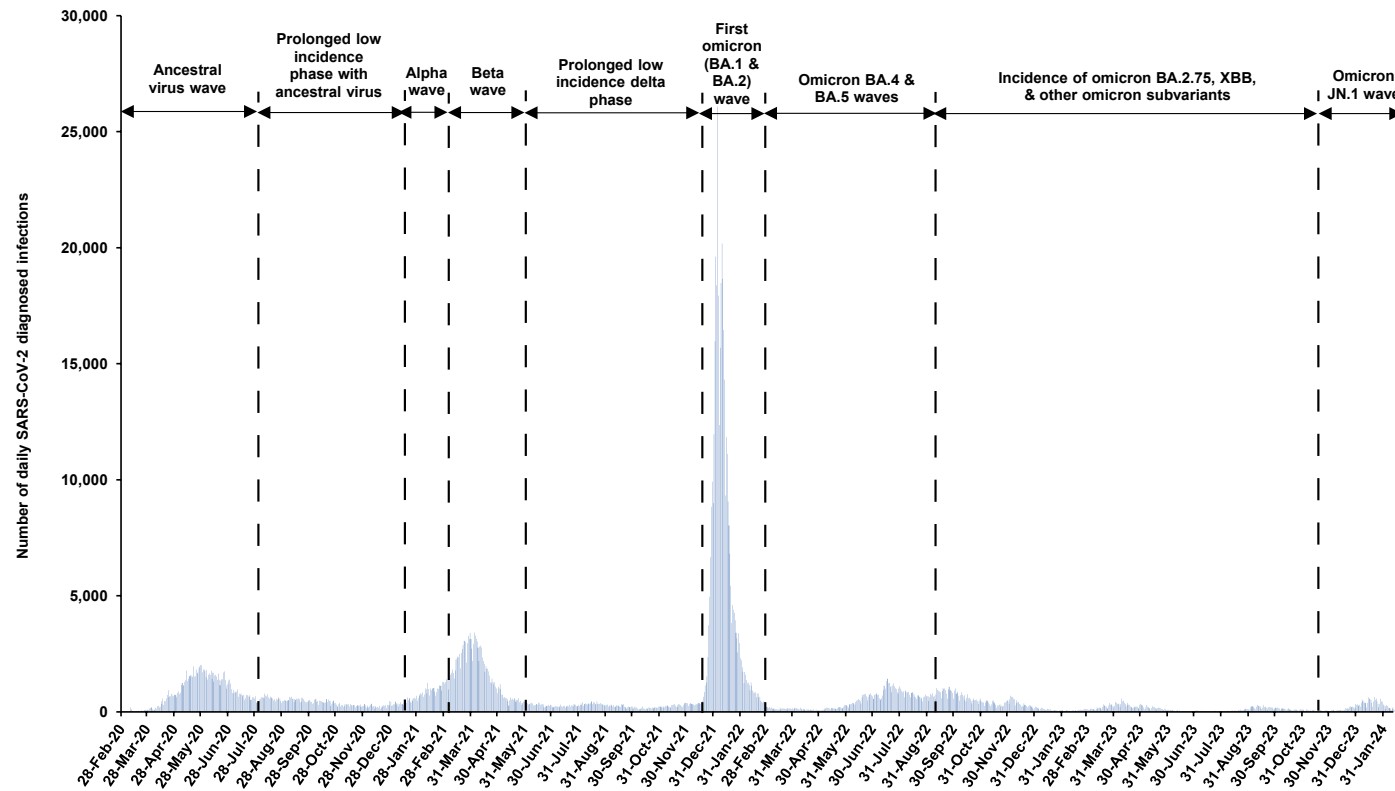

**Extended Data Fig. 3 | SARS-CoV-2 infection incidence in Qatar.** Daily count of newly diagnosed SARS-CoV-2 infections up to the end of the study, between February 5, 2020 and February 12, 2024.

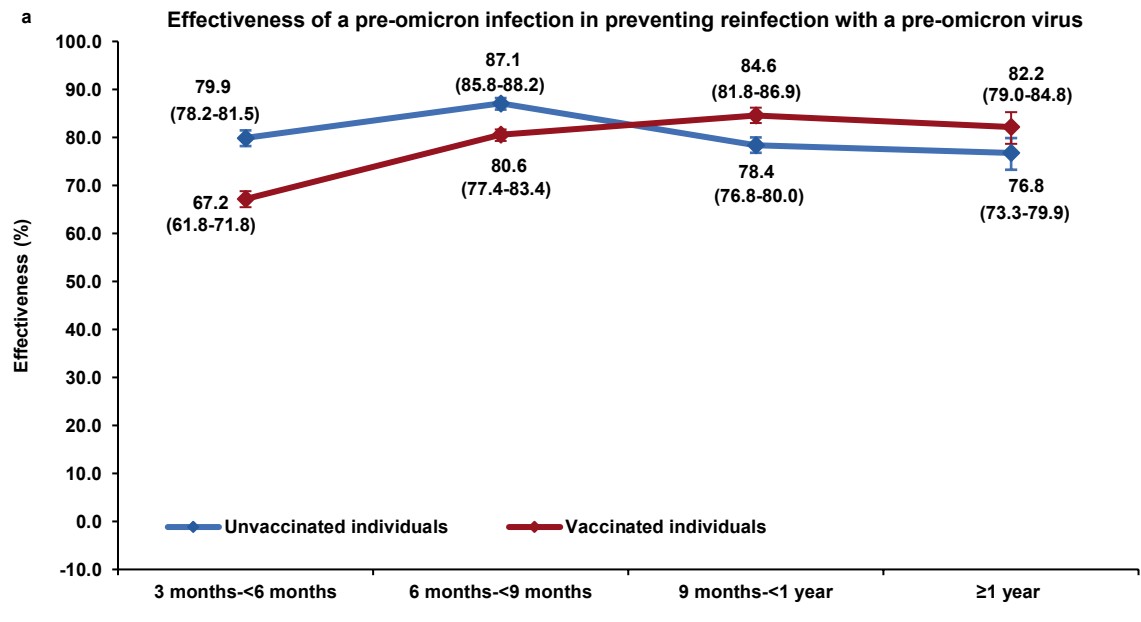

**a** Effectiveness of a pre-omicron infection in preventing reinfection with a pre-omicron virus

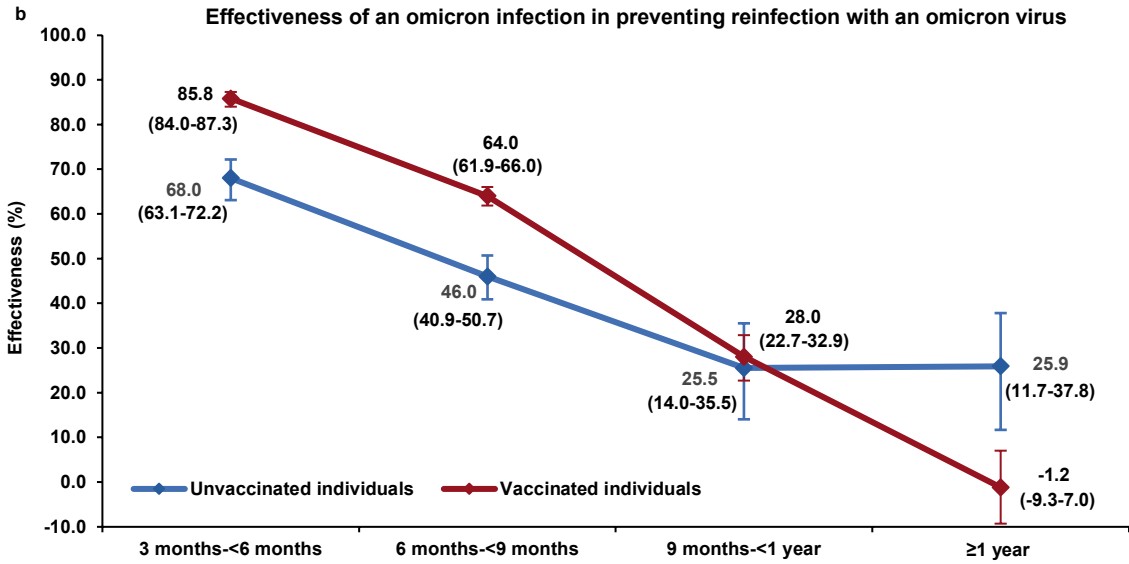

**b** Effectiveness of an omicron infection in preventing reinfection with an omicron virus

**Extended Data Fig. 4 | Effectiveness of previous infection in preventing reinfection by vaccination status.** Effectiveness of infection in preventing reinfection by time since prior infection by vaccination status in the **a**) pre-omicron era and **b**) omicron era. **a** includes 353,485 and 627,836 independent samples for unvaccinated cases and controls, respectively, and 30,746 and 56,421 independent samples for vaccinated cases and controls, respectively.

**b** includes 93,830 and 157,697 independent samples for unvaccinated cases and controls, respectively, and 176,161 and 275,258 independent samples for vaccinated cases and controls, respectively. Data are presented as effectiveness point estimates and corresponding 95% confidence intervals. Error bars indicate the 95% confidence intervals. Measures were not adjusted for multiplicity.

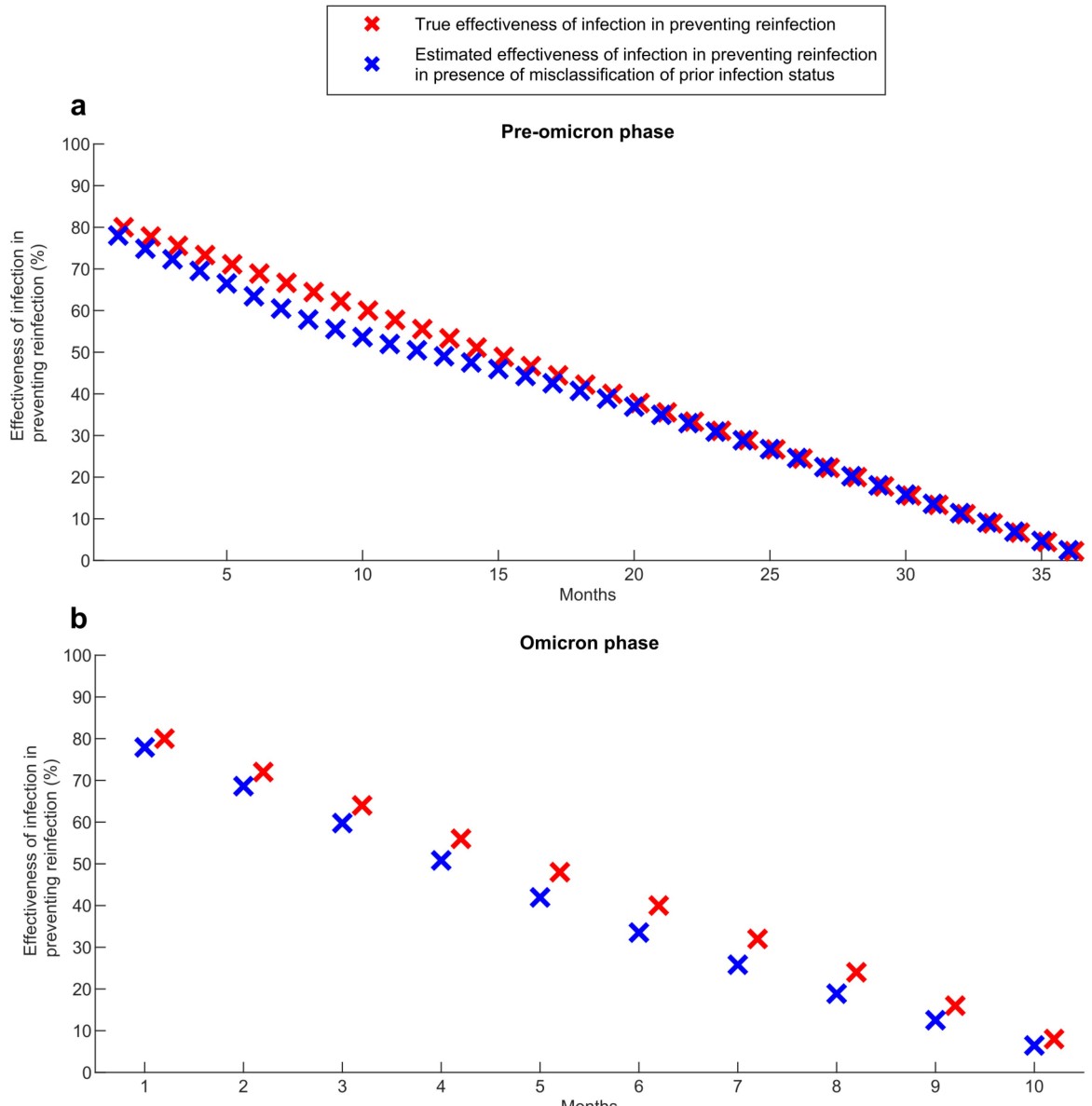

SARS-CoV-2: Severe acute respiratory syndrome coronavirus 2.

**Extended Data Fig. 5 | Impact of bias due to misclassification of prior infection status.** Impact of bias due to misclassification of prior infection status in the test-negative design, assuming 90% of SARS-CoV-2 infections are undocumented. The figure shows the simulated estimated effectiveness of infection against reinfection in presence of misclassification of prior infection status compared to the true effectiveness of infection against reinfection in the pre-omicron **a**) and omicron **b**) phases of the pandemic.

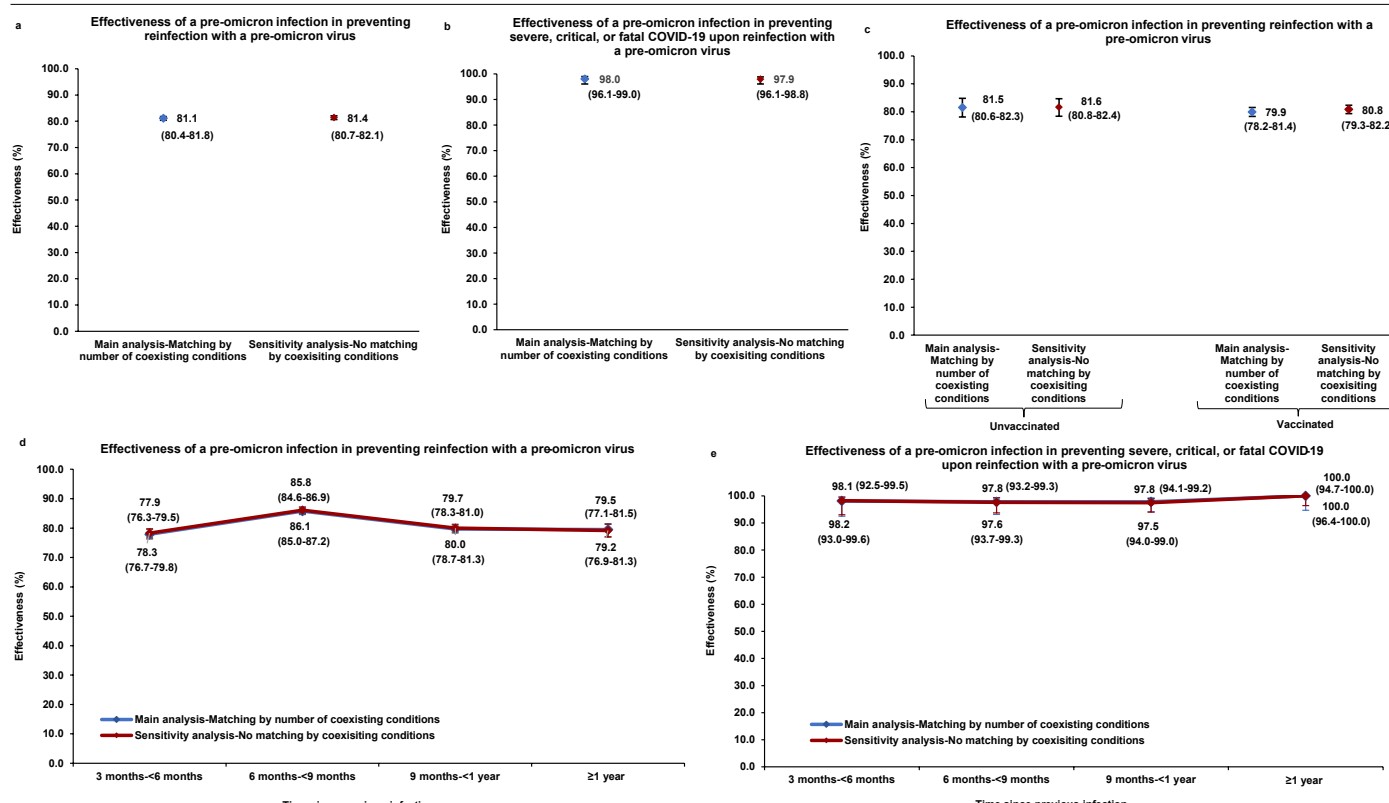

**Extended Data Fig. 6 | Impact of misclassification of coexisting condition status on the results for the pre-omicron era.** The figure compares the main analysis results to those of a sensitivity analysis, where matching by the number of coexisting conditions was entirely removed to simulate complete under-ascertainment of coexisting conditions. The comparisons include: **a**) overall effectiveness of infection in preventing reinfection, **b**) overall effectiveness of infection in preventing severe, critical, or fatal COVID-19 upon reinfection, **c**) overall effectiveness of infection in preventing reinfection for each of vaccinated and unvaccinated individuals, **d**) effectiveness of infection in preventing reinfection by time since prior infection, and **e**) effectiveness of infection in preventing severe, critical, or fatal COVID-19 upon reinfection by time since prior infection. Main analysis includes 384,231 and 684,257 independent samples for cases and controls, respectively. Sensitivity analysis includes 401,686 and 723,529 independent samples for cases and controls, respectively. Data are presented as effectiveness point estimates and corresponding 95% confidence intervals. Error bars indicate the 95% confidence intervals. Measures were not adjusted for multiplicity.

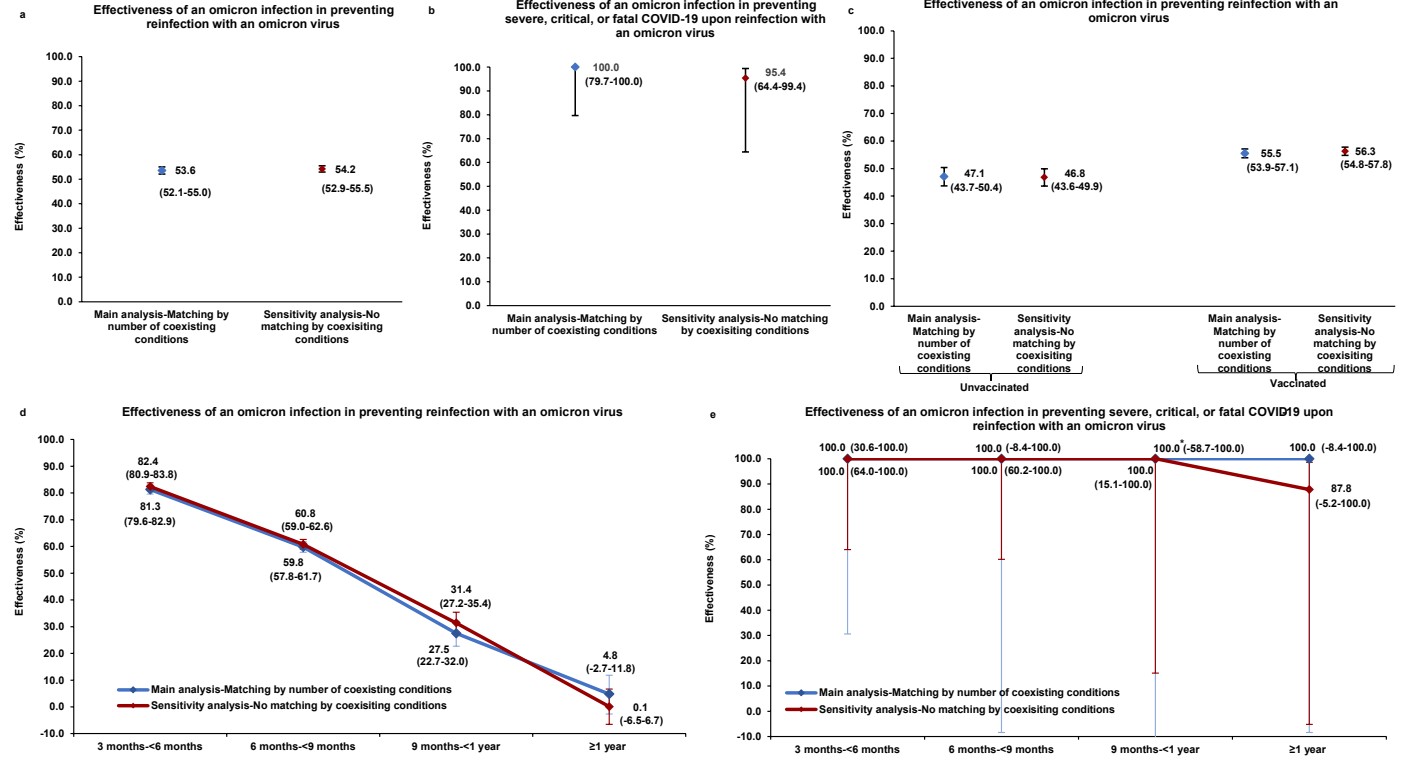

*The negative loswer bound oof the confidence interval was truncated because the confidence interval was too wide.

**Extended Data Fig. 7 | Impact of misclassification of coexisting condition status on the results for the omicron era.** The figure compares the main analysis results to those of a sensitivity analysis, where matching by the number of coexisting conditions was entirely removed to simulate complete under-ascertainment of coexisting conditions. The comparisons include: **a**) overall effectiveness of infection in preventing reinfection, **b**) overall effectiveness of infection in preventing severe, critical, or fatal COVID-19 upon reinfection, **c**) overall effectiveness of infection in preventing reinfection for each of vaccinated and unvaccinated individuals, **d**) effectiveness of infection in

preventing reinfection by time since prior infection, and **e**) effectiveness of infection in preventing severe, critical, or fatal COVID-19 upon reinfection by time since prior infection. Main analysis includes 269,991 and 432,955 independent samples for cases and controls, respectively. Sensitivity analysis includes 287,576 and 464,064 independent samples for cases and controls, respectively. Data are presented as effectiveness point estimates and corresponding 95% confidence intervals. Error bars indicate the 95% confidence intervals. Measures were not adjusted for multiplicity.

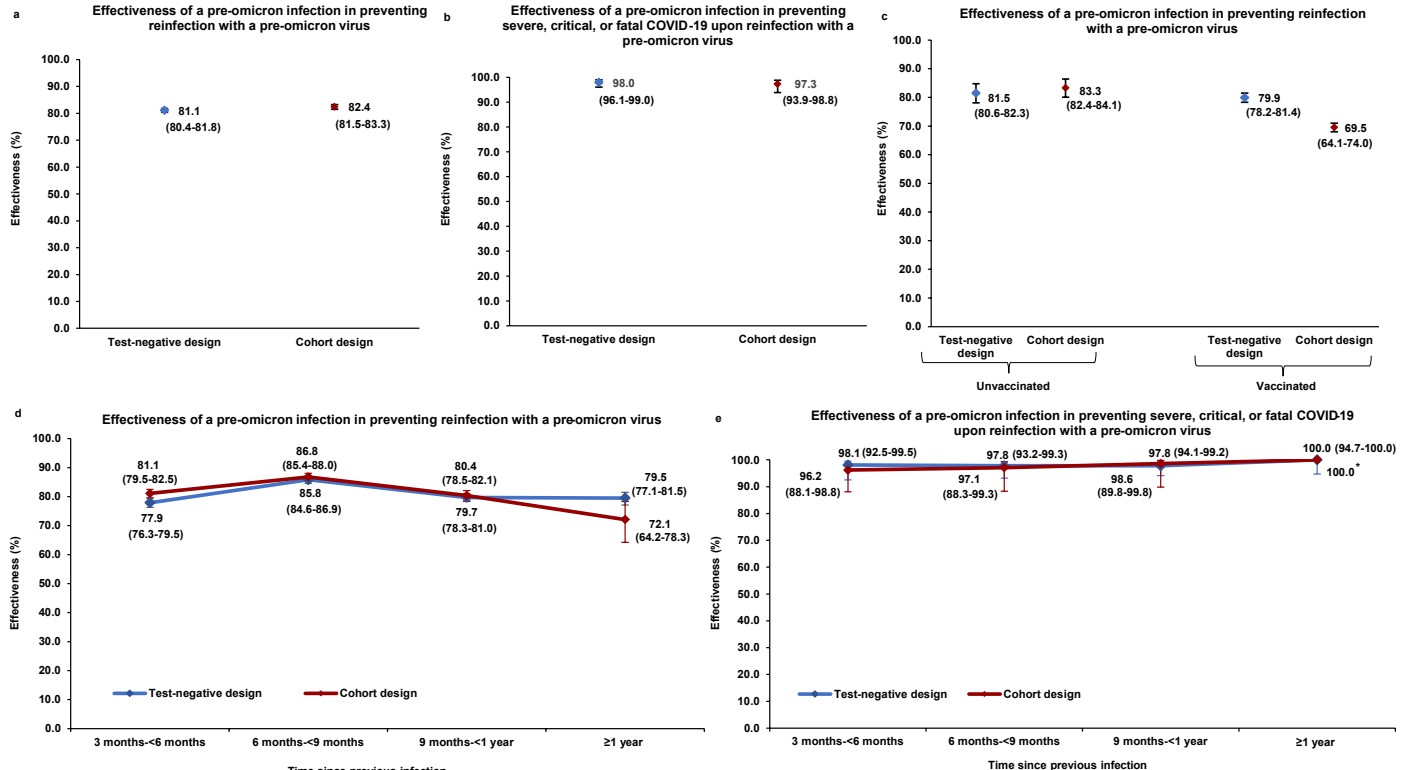

*Confidence interval could not be estimated because zero reinfections progressed to severe critical, or fatal COVID-19 in the primary infection cohort.

**Extended Data Fig. 8 | Validation of the results for the pre-omicron era using a cohort study design.** The figure compares the main analysis results obtained using the test-negative design to the results obtained using the cohort study design. The comparisons include: **a**) overall effectiveness of infection in preventing reinfection, **b**) overall effectiveness of infection in preventing severe, critical, or fatal COVID-19 upon reinfection, **c**) overall effectiveness of infection in preventing reinfection for each of vaccinated and unvaccinated individuals, **d**) effectiveness of infection in preventing reinfection by time since prior infection, and **e**) effectiveness of infection in preventing severe, critical, or fatal COVID-19 upon reinfection by time since prior infection. Analysis using the test-negative design includes 384,231 and 684,257 independent samples for cases and controls, respectively. Analysis using the cohort design includes 339,019 independent samples for each of the primary infection and uninfected cohorts. Data are presented as effectiveness point estimates and corresponding 95% confidence intervals. Error bars indicate the 95% confidence intervals. Measures were not adjusted for multiplicity.

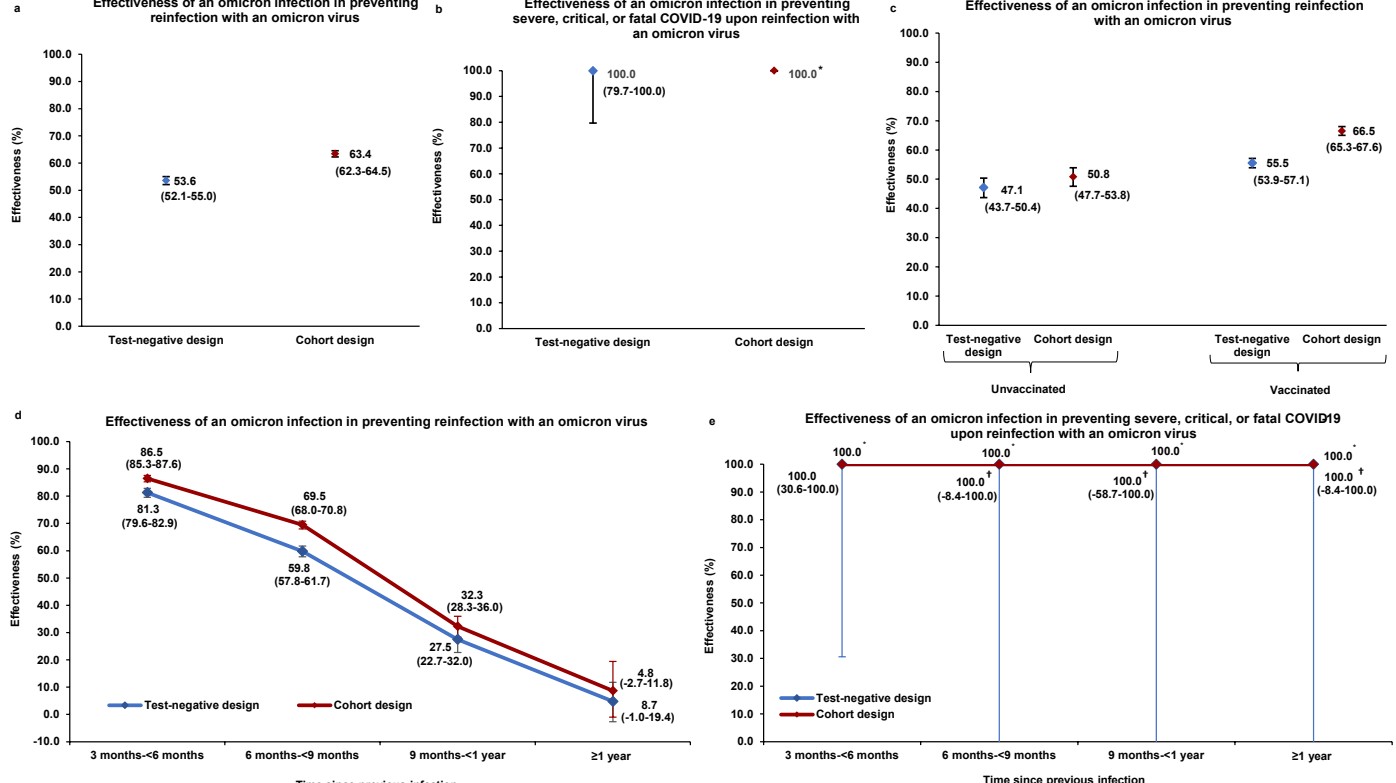

*Confidence interval could not be estimated because zero reinfections progressed to severe critical, or fatal COVID-19 in the primary infection cohort.
†The negative lower bound for the confidence interval was truncated because the confidence interval was too wide.

**Extended Data Fig. 9 | Validation of the results for the omicron era using a cohort study design.** The figure compares the main analysis results obtained using the test-negative design to the results obtained using the cohort study design. The comparisons include: **a**) overall effectiveness of infection in preventing reinfection, **b**) overall effectiveness of infection in preventing severe, critical, or fatal COVID-19 upon reinfection, **c**) overall effectiveness of infection in preventing reinfection for each of vaccinated and unvaccinated individuals, **d**) effectiveness of infection in preventing reinfection by time since prior infection, and **e**) effectiveness of infection in preventing severe, critical, or fatal COVID-19 upon reinfection by time since prior infection. Analysis using the test-negative design includes 269,991 and 432,955 independent samples for cases and controls, respectively. Analysis using the cohort design includes 329,117 independent samples for each of the primary infection and uninfected cohorts. Data are presented as effectiveness point estimates and corresponding 95% confidence intervals. Error bars indicate the 95% confidence intervals. Measures were not adjusted for multiplicity.

**Extended Data Table 1 | Study population characteristics**

| Characteristics | A) Effectiveness of a pre-omicron infection against reinfection with a pre-omicron virus | | | B) Effectiveness of an omicron infection against reinfection with an omicron virus | | |
|---|---|---|---|---|---|---|
| | Cases* | Controls* | SMD† | Cases* | Controls* | SMD† |
| | N=384,231 | N= 684,257 | | N=269,991 | N=432,955 | |
| **Median age (IQR) — years** | 32 (25-40) | 32 (25-39) | 0.03‡ | 33 (24-40) | 33 (24-40) | 0.01‡ |
| **Age group — no. (%)** | | | | | | |
| <10 years | 31,211 (8.1) | 58,112 (8.5) | | 23,605 (8.7) | 37,538 (8.7) | |
| 10-19 years | 27,854 (7.2) | 50,909 (7.4) | | 24,471 (9.1) | 38,643 (8.9) | |
| 20-29 years | 91,384 (23.8) | 165,438 (24.2) | | 58,777 (21.8) | 95,917 (22.2) | |
| 30-39 years | 134,865 (35.1) | 241,466 (35.3) | 0.03 | 90,469 (33.5) | 146,060 (33.7) | 0.02 |
| 40-49 years | 68,349 (17.8) | 118,002 (17.2) | | 47,877 (17.7) | 76,781 (17.7) | |
| 50-59 years | 23,923 (6.2) | 39,703 (5.8) | | 18,327 (6.8) | 28,439 (6.6) | |
| 60-69 years | 5,477 (1.4) | 8,731 (1.3) | | 4,934 (1.8) | 7,463 (1.7) | |
| 70+ years | 1,168 (0.3) | 1,896 (0.3) | | 1,531 (0.6) | 2,114 (0.5) | |
| **Sex** | | | | | | |
| Male | 277,586 (72.2) | 488,035 (71.3) | 0.02 | 163,503 (60.6) | 268,826 (62.1) | 0.03 |
| Female | 106,645 (27.8) | 196,222 (28.7) | | 106,488 (39.4) | 164,129 (37.9) | |
| **Nationality§** | | | | | | |
| Bangladeshi | 31,612 (8.2) | 52,824 (7.7) | | 10,551 (3.9) | 18,632 (4.3) | |
| Egyptian | 21,622 (5.6) | 38,164 (5.6) | | 9,065 (3.4) | 13,293 (3.1) | |
| Filipino | 36,523 (9.5) | 66,857 (9.8) | | 32,088 (11.9) | 48,608 (11.2) | |
| Indian | 108,354 (28.2) | 192,160 (28.1) | | 64,750 (24.0) | 110,646 (25.6) | |
| Nepalese | 39,806 (10.4) | 67,487 (9.9) | 0.05 | 15,018 (5.6) | 26,123 (6.0) | 0.06 |
| Pakistani | 19,647 (5.1) | 34,479 (5.0) | | 9,714 (3.6) | 16,537 (3.8) | |
| Qatari | 50,206 (13.1) | 98,638 (14.4) | | 64,335 (23.8) | 99,286 (22.9) | |
| Sri Lankan | 11,987 (3.1) | 20,156 (2.9) | | 6,704 (2.5) | 10,648 (2.5) | |
| Sudanese | 9,883 (2.6) | 17,862 (2.6) | | 5,350 (2.0) | 7,671 (1.8) | |
| Other nationalities‖ | 54,591 (14.2) | 95,630 (14.0) | | 52,416 (19.4) | 81,511 (18.8) | |
| **Coexisting conditions¶** | | | | | | |
| 0 | 315,970 (82.2) | 566,661 (82.8) | | 22,7962 (84.4) | 371,528 (85.8) | |
| 1 | 41,525 (10.8) | 72,864 (10.6) | | 25,268 (9.4) | 37,628 (8.7) | |
| 2 | 15,531 (4.0) | 26,322 (3.8) | | 9,151 (3.4) | 13,207 (3.1) | |
| 3 | 5,451 (1.4) | 9,022 (1.3) | 0.02 | 3,428 (1.3) | 4,850 (1.1) | 0.04 |
| 4 | 2,665 (0.7) | 4,315 (0.6) | | 1,774 (0.7) | 2,458 (0.6) | |
| 5 | 1,401 (0.4) | 2,206 (0.3) | | 1,058 (0.4) | 1,449 (0.3) | |
| 6+ | 1,688 (0.4) | 2,867 (0.4) | | 1,350 (0.5) | 1,835 (0.4) | |
| **Vaccine doses¶** | | | | | | |
| 0 | 353,485 (92.0) | 627,836 (91.8) | | 93,830 (34.8) | 157,697 (36.4) | |
| 1 | 11,082 (2.9) | 18,769 (2.7) | | 1,336 (0.5) | 2,006 (0.5) | |
| 2 | 19,510 (5.1) | 37,377 (5.5) | 0.02 | 137,771 (51.0) | 211,640 (48.9) | 0.04 |
| 3 | 154 (0.04) | 275 (0.04) | | 36,991 (13.7) | 61,529 (14.2) | |
| 4 | -- | -- | | 63 (0.0) | 83 (0.0) | |
| **Median time between last vaccine dose and SARS-CoV-2 test (IQR)** | 32 (10-143) | 36 (14-139) | 0.00‡ | 232 (153-290) | 235 (166-287) | 0.02‡ |
| **Method of testing** | | | | | | |
| PCR | 384,231 (100.0) | 684,257 (100.0) | -- | 215,470 (79.8) | 341,172 (78.8) | 0.02 |
| RA | -- | -- | | 54,521 (20.2) | 91,783 (21.2) | |
| **Reason for testing** | | | | | | |
| Clinical suspicion | 140,004 (36.4) | 220,070 (32.2) | | 52,582 (19.5) | 70,299 (16.2) | |
| Contact tracing | 73,350 (19.1) | 134,514 (19.7) | | 40,479 (15.0) | 60,527 (14.0) | |
| Survey | 66,249 (17.2) | 127,545 (18.6) | | 24,526 (9.1) | 39,356 (9.1) | |
| Port of entry | 54,257 (14.1) | 107,762 (15.7) | 0.09 | 41,842 (15.5) | 81,667 (18.9) | 0.12 |
| Individual request | 18,360 (4.8) | 35,606 (5.2) | | 28,979 (10.7) | 46,626 (10.8) | |
| Healthcare routine testing | 21,923 (5.7) | 39,039 (5.7) | | 6,383 (2.4) | 9,830 (2.3) | |
| Pre-travel | 10,088 (2.6) | 19,721 (2.9) | | 75,200 (27.9) | 124,650 (28.8) | |

IQR, interquartile range; PCR polymerase chain reaction; RA rapid antigen; SARS-CoV-2, severe acute respiratory syndrome coronavirus 2; SMD standardized mean difference.

*Cases (SARS-CoV-2-positive tests) and controls (SARS-CoV-2-negative tests) were matched exactly one-to-two by sex, 10-year age group, nationality, number of coexisting conditions, number of vaccine doses at time of the SARS-CoV-2 test, calendar week of the SARS-CoV-2 test, method of testing, and reason for testing.

†SMD is the difference in the mean of a covariate between groups divided by the pooled standard deviation. An SMD of ≤0.1 indicates adequate matching.

‡SMD is for the mean difference between groups divided by the pooled standard deviation.

§Nationalities were chosen to represent the most populous groups in Qatar.

‖These comprise up to 131 other nationalities in Qatar in the analysis for the effectiveness of a pre-omicron infection against reinfection with a pre-omicron virus and up to 135 nationalities in the analysis for the effectiveness of an omicron infection against reinfection with an omicron virus.

¶Ascertained at the time of the SARS-CoV-2 test.

Characteristics of matched cases and controls in samples used to estimate the effectiveness of **a**) a pre-omicron infection in preventing reinfection with a pre-omicron virus and **b**) an omicron infection in preventing reinfection with an omicron virus.

**Extended Data Table 2 | Sensitivity analysis employing 40-day time window to define reinfection**

| Effectiveness | Cases[*] | | Controls[*] | | Effectiveness[†] in % (95% CI)[‡] |
|---|---|---|---|---|---|
| | Previous infection (n) | No previous infection (n) | Previous infection (n) | No previous infection (n) | |
| **a) Effectiveness of a pre-omicron infection against reinfection with a pre-omicron virus** | | | | | |
| Any previous infection | 4,706 | 381,635 | 32,425 | 656,178 | 75.2 (74.4 to 76.0) |
| By time since previous infection | | | | | |
| Subgroup analysis 1 | | | | | |
| 3 months-<6 months | 919 | 378,184 | 7,324 | 655,304 | 77.6 (76.0 to 79.2) |
| 6 months-<9 months | 693 | 378,594 | 8,361 | 655,307 | 85.8 (84.6 to 86.9) |
| 9 months-<1 year | 1,000 | 378,965 | 8,076 | 655,442 | 79.8 (78.4 to 81.1) |
| ≥1 year | 382 | 377,682 | 3,224 | 655,157 | 79.0 (76.6 to 81.2) |
| Subgroup analysis 2 | | | | | |
| <1 year | 4,314 | 381,468 | 29,185 | 656,141 | 74.8 (74.0 to 75.6) |
| ≥1 year | 382 | 377,682 | 3,224 | 655,157 | 79.0 (76.6 to 81.2) |
| **b) Effectiveness of an omicron infection against reinfection with an omicron virus** | | | | | |
| Any previous infection | 7,882 | 262,746 | 24,489 | 409,753 | 56.3 (54.9 to 57.6) |
| By time since previous infection | | | | | |
| Subgroup analysis 1 | | | | | |
| 3 months-<6 months | 648 | 258,814 | 4,845 | 406,860 | 80.9 (79.1 to 82.5) |
| 6 months-<9 months | 2,622 | 259,707 | 9,606 | 407,673 | 59.9 (57.9 to 61.9) |
| 9 months-<1 year | 1,914 | 258,545 | 4,077 | 407,357 | 27.1 (22.3 to 31.6) |
| ≥1 year | 1,976 | 258,306 | 2,922 | 407,345 | 1.6 (-6.0 to 9.0) |
| Subgroup analysis 2 | | | | | |
| <1 year | 5,697 | 262,044 | 21,320 | 408,933 | 61.8 (60.5 to 63.1) |
| ≥1 year | 1,976 | 258,306 | 2,922 | 407,345 | 1.6 (-6.0 to 9.0) |

CI, confidence interval; SARS-CoV-2, severe acute respiratory syndrome coronavirus 2.

[*]Cases (SARS-CoV-2-positive tests) and controls (SARS-CoV-2-negative tests) were matched exactly one-to-two by sex, 10-year age group, nationality, number of coexisting conditions, number of vaccine doses at time of the SARS-CoV-2 test, calendar week of the SARS-CoV-2 test, method of testing, and reason for testing.

[†]Effectiveness of previous infection in preventing reinfection was estimated using the test-negative, case–control study design.

[‡]CIs were not adjusted for multiplicity and thus should not be used to infer definitive differences between different groups.

Effectiveness of a **a)** pre-omicron infection in preventing reinfection with a pre-omicron virus and **b)** an omicron infection in preventing reinfection with an omicron virus, assuming a 40-day time window for defining reinfection instead of the conventional 90-day time window.

## Extended Data Table 3 | Reporting checklist

| | Item No | Recommendation | Main text page |
|---|---|---|---|
| **Title and abstract** | 1 | (*a*) Indicate the study's design with a commonly used term in the title or the abstract | Summary |
| | | (*b*) Provide in the abstract an informative and balanced summary of what was done and what was found | Summary |
| **Introduction** | | | |
| Background/ rationale | 2 | Explain the scientific background and rationale for the investigation being reported | Introduction |
| Objectives | 3 | State specific objectives, including any prespecified hypotheses | Introduction |
| **Methods** | | | |
| Study design | 4 | Present key elements of study design | Methods ('Study population and data sources' & 'Study design') |
| Setting | 5 | Describe the setting, locations, and relevant dates, including periods of recruitment, exposure, follow-up, and data collection | Methods ('Study population and data sources' & 'Study design'), & Supplementary Information Section 2, & Extended Data Figs. 1 & 2 |
| Participants | 6 | (*a*) Give the eligibility criteria, and the sources and methods of case ascertainment and control selection. Give the rationale for the choice of cases and controls; (*b*) For matched studies, give matching criteria and the number of controls per case | Methods ('Study population and data sources' & 'Study design'), & Supplementary Information Section 2, & Extended Data Figs. 1 & 2 |
| Variables | 7 | Clearly define all outcomes, exposures, predictors, potential confounders, and effect modifiers. Give diagnostic criteria | Methods ('Study design', & 'Statistical Analysis'), Extended Data Table 1, & Supplementary Information Sections 1-4 |
| Data sources/ measurement | 8 | For each variable of interest, give sources of data and details of methods of assessment (measurement). Describe comparability of assessment methods if there is more than one group | Methods ('Study population and data sources', 'Study design', & 'Statistical analysis') & Supplementary Information Sections 1-4 |
| Bias | 9 | Describe any efforts to address potential sources of bias | Methods ('Study design' & 'Statistical analysis') |
| Study size | 10 | Explain how the study size was arrived at | Extended Data Figs. 1 & 2 |
| Quantitative variables | 11 | Explain how quantitative variables were handled in the analyses. If applicable, describe which groupings were chosen and why | Methods ('Study design' & 'Statistical analysis'), Extended Data Table 1, & Supplementary Information Sections 1 & 3-4 |
| Statistical methods | 12 | (*a*) Describe all statistical methods, including those used to control for confounding; (*b*) Describe any methods used to examine subgroups and interactions | Methods ('Statistical analysis') |
| | | (*c*) Explain how missing data were addressed | Not applicable, see Methods ('Study population and data sources') & Supplementary Information Section 2 |
| | | (*d*) If applicable, explain how matching of cases and controls was addressed | Methods ('Study design' & 'Statistical analysis') |
| | | (*e*) Describe any sensitivity analyses | Methods ('Statistical analysis' & 'Additional validation analyses') |
| **Results** | | | |
| Participants | 13 | (a) Report numbers of individuals at each stage of study—eg numbers potentially eligible, examined for eligibility, confirmed eligible, included in the study, completing follow-up, and analysed; (b) Give reasons for non-participation at each stage; (c) Consider use of a flow diagram | Extended Data Figs. 1 & 2 |
| Descriptive data | 14 | (a) Give characteristics of study participants (eg demographic, clinical, social) and information on exposures and potential confounders | 'Study populations' & Extended Data Table 1 |
| | | (b) Indicate number of participants with missing data for each variable of interest | Not applicable, see Methods ('Study population and data sources') & Supplementary Information Section 2 |
| Outcome data | 15 | Report numbers in each exposure category, or summary measures of exposure | 'Pre-omicron immunity against pre-omicron' & 'Omicron immunity against omicron', Table 1, & Fig. 1 |
| Main results | 16 | (*a*) Give unadjusted estimates and, if applicable, confounder-adjusted estimates and their precision (eg, 95% confidence interval). | 'Pre-omicron immunity against pre-omicron' & 'Omicron immunity against omicron', Table 1, & Fig. 1 |
| | | (*b*) Report category boundaries when continuous variables were categorized | Extended Data Table 1 |
| | | (*c*) If relevant, consider translating estimates of relative risk into absolute risk for a meaningful time period | Not applicable |
| Other analyses | 17 | Report other analyses done—eg analyses of subgroups and interactions, and sensitivity analyses | 'Pre-omicron immunity against pre-omicron', 'Omicron immunity against omicron', & 'Additional validation analyses', Tables 2 & 3, Extended Data Table 2, & Extended Data Figs 4-9 |
| **Discussion** | | | |
| Key results | 18 | Summarise key results with reference to study objectives | Discussion, paragraphs 1-8 |
| Limitations | 19 | Discuss limitations of the study, taking into account sources of potential bias or imprecision. | Discussion, paragraphs 9-14 |
| Interpretation | 20 | Give a cautious overall interpretation of results considering objectives, limitations, multiplicity of analyses, results from similar studies, and other relevant evidence | Discussion, paragraph 16 |
| Generalisability | 21 | Discuss the generalisability (external validity) of the study results | Discussion, paragraph 12 |
| **Other information** | | | |
| Funding | 22 | Give source of funding and role of funders for the present study | Acknowledgements |

Strengthening the Reporting of Observational Studies in Epidemiology (STROBE) checklist for case-control studies.

# Reporting Summary

## Statistics

For all statistical analyses, confirm that the following items are present in the figure legend, table legend, main text, or Methods section.

| n/a | Confirmed | |
|---|---|---|
| ☐ | ☒ | The exact sample size (*n*) for each experimental group/condition, given as a discrete number and unit of measurement |
| ☐ | ☒ | A statement on whether measurements were taken from distinct samples or whether the same sample was measured repeatedly |
| ☒ | ☐ | The statistical test(s) used AND whether they are one- or two-sided *Only common tests should be described solely by name; describe more complex techniques in the Methods section.* |
| ☐ | ☒ | A description of all covariates tested |
| ☐ | ☒ | A description of any assumptions or corrections, such as tests of normality and adjustment for multiple comparisons |
| ☐ | ☒ | A full description of the statistical parameters including central tendency (e.g. means) or other basic estimates (e.g. regression coefficient) AND variation (e.g. standard deviation) or associated estimates of uncertainty (e.g. confidence intervals) |
| ☒ | ☐ | For null hypothesis testing, the test statistic (e.g. *F*, *t*, *r*) with confidence intervals, effect sizes, degrees of freedom and *P* value noted *Give P values as exact values whenever suitable.* |
| ☒ | ☐ | For Bayesian analysis, information on the choice of priors and Markov chain Monte Carlo settings |
| ☒ | ☐ | For hierarchical and complex designs, identification of the appropriate level for tests and full reporting of outcomes |
| ☐ | ☒ | Estimates of effect sizes (e.g. Cohen's *d*, Pearson's *r*), indicating how they were calculated |

*Our web collection on statistics for biologists contains articles on many of the points above.*

## Software and code

Policy information about availability of computer code

| Data collection | Data were entered into the CERNER database system. |
|---|---|
| Data analysis | Standard epidemiological analyses were conducted using standard commands in STATA/SE 18.0. These commands have been published at: https://github.com/IDEGWCMQ/TestNegCode/blob/main/TestNegCode.txt |

For manuscripts utilizing custom algorithms or software that are central to the research but not yet described in published literature, software must be made available to editors and reviewers. We strongly encourage code deposition in a community repository (e.g. GitHub). See the Nature Portfolio guidelines for submitting code & software for further information.

## Data

Policy information about availability of data

All manuscripts must include a data availability statement. This statement should provide the following information, where applicable:
- Accession codes, unique identifiers, or web links for publicly available datasets
- A description of any restrictions on data availability
- For clinical datasets or third party data, please ensure that the statement adheres to our policy

The National Coronavirus Disease 2019 (COVID-19) dataset used in this study is a property of the Qatar Ministry of Public Health that was provided to the researchers through a restricted-access agreement that prevents sharing the dataset with a third party or publicly. This dataset encompasses the National COVID-19 Testing Database, the National COVID-19 Vaccination Database, the National COVID-19 Severity Database, and the National Mortality Database. These data are

# Research involving human participants, their data, or biological material

Policy information about studies with human participants or human data. See also policy information about sex, gender (identity/presentation), and sexual orientation and race, ethnicity and racism.

| | |
|---|---|
| Reporting on sex and gender | The study populations are balanced by sex (please see Extended Data Table 1). Sex is as recorded in the integrated nationwide digital-health information platform, which is based on the Qatar Identity Card. |
| Reporting on race, ethnicity, or other socially relevant groupings | The study populations are balanced across nationality groups (please see Extended Data Table 1). Nationality, age, and sex provide a powerful proxy for occupation and socio-economic status in Qatar as evidenced by earlier studies in this population. |
| Population characteristics | The demographic characteristics of the study populations can be found in Extended Data Table 1. |
| Recruitment | This is a test-negative case control study where the odds of previous infection was compared between cases (defined as SARS-CoV-2-positive tests) and controls (defined as SARS-CoV-2-negative tests) before and after the the introduction of the omicron variant in Qatar on December 19, 2021. Two analyses were conducted. The first analysis was conducted between February 5, 2020 (the onset of the COVID-19 pandemic in Qatar), and December 18, 2021 to assess the effectiveness of a pre-omicron infection in preventing reinfection with a pre-omicron virus. The second analysis was conducted between December 19, 2021, and February 12, 2024 (marking the end of the study) to assess the effectiveness of an omicron infection in preventing reinfection with an omicron virus. COVID-19 laboratory testing, clinical infection data, severity, hospitalization, vaccination, and related demographic details were extracted from the integrated nationwide digital-health information platform that hosts the national, federated SARS-CoV-2 databases. These databases are complete with no missing information for PCR testing, medically-supervised rapid antigen testing, COVID-19 vaccinations, COVID-19 hospitalizations and deaths, and basic demographic details, and have captured all SARS-CoV-2-related data since epidemic onset. This study was conducted on the entire resident population of Qatar. SARS-CoV-2 testing was extensive in Qatar until October 31, 2022, with nearly 5% of the population being tested every week, primarily for routine purposes such as screening or meeting travel-related requirements. Subsequently, testing rates decreased, with less than 1% of the population being tested per week. The majority of infections during the pandemic were diagnosed through routine testing rather than symptomatic presentation. |
| Ethics oversight | The study was approved by the Hamad Medical Corporation and Weill Cornell Medicine-Qatar Institutional Review Boards with waiver of informed consent. |

Note that full information on the approval of the study protocol must also be provided in the manuscript.

# Field-specific reporting

Please select the one below that is the best fit for your research. If you are not sure, read the appropriate sections before making your selection.

☒ Life sciences   ☐ Behavioural & social sciences   ☐ Ecological, evolutionary & environmental sciences

For a reference copy of the document with all sections, see nature.com/documents/nr-reporting-summary-flat.pdf

# Life sciences study design

All studies must disclose on these points even when the disclosure is negative.

| | |
|---|---|
| Sample size | COVID-19 laboratory testing, clinical infection data, severity, hospitalization, vaccination, and related demographic details were extracted from the integrated nationwide digital-health information platform that hosts the national, federated SARS-CoV-2 databases. These databases are complete and have captured all SARS-CoV-2-related data since epidemic onset. The data is for the entire national population and includes every individual tested for SARS-CoV-2 in Qatar. The sample size varied depending on the definition used for cases (SARS-CoV-2-positive tests, as well as severe, critical, or fatal COVID-19 due to a SARS-CoV-2 infection), and controls (SARS-CoV-2-negative tests), during each of the pre-omicron and omicron phases of the pandemic. Cases and controls were matched exactly one-to-two by sex, 10-year age group, nationality, number of coexisting conditions, number of vaccine doses, calendar week of the SARS-CoV-2 test, method of testing (PCR or rapid antigen), and reason for testing in estimating the effectiveness of previous infection in preventing reinfection. A one-to-five matching ratio was applied in estimating effectiveness in preventing severe, critical, or fatal COVID-19 upon reinfection, to enhance statistical precision. In each analysis for a specific time-since-previous-infection stratum, we included only those with previous infection in this specific time-since-previous-infection stratum and those with no previous infection (our reference group). Thus, the number of cases (and controls) varied across time-since-previous-infection analyses. Given that the sample sizes were based on the entire national population with only individuals that do not fit the eligibility criteria excluded, the sample size for each sub-study can be considered sufficient. Detailed sample sizes can be found in Extended Data Figs. 1 and 2, Extended Data Table 1, and Table 1. |
| Data exclusions | Exclusion criteria were specified a priori. SARS-CoV-2 reinfection is conventionally defined as a documented infection ≥90 days after a previous infection, to avoid misclassifying prolonged test positivity as a reinfection with shorter time intervals. Consequently, cases or controls preceded by SARS-CoV-2-positive tests within 90 days were excluded. To comply with the non-differential healthcare-seeking behavior assumption inherent to the test-negative study design, only tests with a documented reason for testing were included in the analysis. In the |

omicron-era analysis, cases or controls preceded by a pre-omicron infection were excluded from the analysis, as the research question pertained to the effectiveness of omicron immunity, rather than pre-omicron immunity, against omicron reinfection.

| | |
|---|---|
| Replication | To validate the results and ensure reproducibility, six additional independent analyses were conducted to estimate the effectiveness of previous infection in preventing reinfection overall and by time since previous infection after 1) restricting the analysis to tests performed due to clinical suspicion, indicating the presence of symptoms consistent with a respiratory tract infection, 2) stratifying the analysis by vaccination status, 3) redefining SARS-CoV-2 reinfection as a documented infection occurring ≥40 days after a previous infection, instead of the conventional ≥90 days, 4) removing the requirement of matching by the number of coexisting conditions, 5) using a cohort study design to estimate natural infection protection instead of a test-negative design, and 6) using mathematical modeling to evaluate the impact of misclassification of prior infection status on the estimated waning pattern of immune protection. All analyses confirmed/reproduced estimates of the effectiveness of previous infection against reinfection obtained in the main analysis. |
| Randomization | Not applicable as this is an observational case-control study where individuals are aware of their infection status. However, controls were selected from the entire national population, and exact matching on multiple factors was employed to ensure rigorous pairing of cases and controls. To further ensure that effectiveness estimates were not biased, the study design controlled for vaccination status, allowing for differentiation between the effects of prior infection and vaccination, another strength of the test-negative design. |
| Blinding | Not applicable as this is an observational case-control study where individuals are aware of their infection status. |

# Reporting for specific materials, systems and methods

We require information from authors about some types of materials, experimental systems and methods used in many studies. Here, indicate whether each material, system or method listed is relevant to your study. If you are not sure if a list item applies to your research, read the appropriate section before selecting a response.

## Materials & experimental systems

| n/a | Involved in the study |
|---|---|
| ☒ | Antibodies |
| ☒ | Eukaryotic cell lines |
| ☒ | Palaeontology and archaeology |
| ☒ | Animals and other organisms |
| ☒ | Clinical data |
| ☒ | Dual use research of concern |
| ☒ | Plants |

## Methods

| n/a | Involved in the study |
|---|---|
| ☒ | ChIP-seq |
| ☒ | Flow cytometry |
| ☒ | MRI-based neuroimaging |

