## [Peer Review file · Nature]

Differential Protection Against SARS-CoV-2 Reinfection Pre- and Post-Omicron

Corresponding Author: Professor Laith Abu-Raddad

Version 0:

Reviewer comments:

Referee #1

(Remarks to the Author)

This manuscript delivers important epidemiological and statistical analyses showing that pre-Omicron and Omicron infections exhibit distinct efficacies of protection against subsequent reinfections. Specifically, the protection induced by Omicron infection decreases significantly faster than the protection provided by pre-Omicron infection. The results underscore the importance of updating SARS-CoV-2 vaccines through clinical evidence, an aspect that should be appreciated and will interest those in this field, although this lacks discussion in the manuscript.

However, the accelerated evolution of Omicron lineages due to the immune pressure from neutralizing antibodies, highlighted by the authors as a potential explanation for the observation, has already been investigated at the molecular level and convincing evidence has been obtained in previous publications. And, the study lacks in-depth discussions of the potential relationship and interaction between the observed phenomenon, virus evolution, and the potential contribution of humoral immune imprinting. Despite its significance, I would not recommend this manuscript for publication in Nature.

Major problem:

The major conclusion proposed by the author is already well-accepted in the field. Many virological analyses have obtained direct evidence regarding the difference in evolution tendency between the pre and post-Omicron eras. Other factors may also contribute to the observation but lack discussions, such as:

1. It could result from immune imprinting. Due to the common ancestral strain vaccination history in the population, Omicron infection, which is antigenically distinct compared to the ancestral strain, fails to elicit sufficient Omicron-neutralizing antibodies.
2. It could also be attributed to the rapid convergent evolution of the SARS-CoV-2 RBD.

Despite I do enjoy the concise concept and results in this manuscript, I believe it would be more suitable to publish in a specialist journal.

Referee #3

(Remarks to the Author)

Chemaitelly et al. conducted a test-negative study investigating differential effectiveness particularly durability of previous infection with pre-Omicron SARS-CoV-2 variants in protection against pre-Omicron reinfections in comparison with protection of previous Omicron infection against Omicron reinfection using a population-based database of individuals who received a RT-PCR test for SARS-CoV-2 in the country during 5 Feb 2020-12 Feb 2024.

It's not surprising to see the study conclusion "These patterns suggest a shift in evolutionary pressures, with intrinsic transmissibility driving adaptation pre-omicron and immune escape becoming dominant post-omicron", which has already been shown in previous virologic and human studies (Carabelli, Nat Rev Microbiol 2023; Willet, Nat Rev Microbiol 2022), in addition to our general understanding of protection that tends to be long-lasting against infection with same/more similar strains.

A series of studies have been published examining protection against reinfection/breakthrough infection with Omicron/pre-Omicron variants provided by previous infection/vaccination using the same database but perhaps from different periods.

More technical details are needed to determine the validity of the study findings.

1. Since the study period extends to Feb 2024 when the COVID-19 pandemic had been announced to end, I suspect the ascertainment of Omicron infections was substantially lower than that during the pre-Omicron period especially as explained in the Supplementary information that RATs were gradually increasing during Omicron era. How much would this affect the results of the analysis and therefore the interpretation of the findings?

2. Although the authors acknowledged that potential biases and confounders which might have not been accounted for sufficiently are the challenges faced in the real-world data analyses, it would be difficult to justify the validity of the analyses without further information on how much the current data might have been affected/ not affected by those potential biases such as non-binary healthcare seeking behaviors, testing behaviors/accessibility (to RT-PCR), timing of vaccination in addition to doses of vaccination, etc. As indicated in lines 201-208, the counterintuitive higher estimates of the effectiveness against any reinfection than that for symptomatic infections were explained by the differential timing between previous infection and reinfection. It would be more informative to have supporting data to verify how much the analyses would be affected by potential biases/confounders.

3. Supplementary Table 1, why is the overall estimate of the effectiveness of pre-Omicron infection against reinfection of pre-Omicron variants lower than any of the estimates from the subgroup analyses, a similar question for the estimate for <1 year in subgroup analysis 2?

4. As described, comorbidities were defined based on the records in the public healthcare system. How much do the authors think the under-ascertainment of information on those whose comorbidities were not captured by the database would affect their results? Any supporting data for that?

5. As they previously showed (Chemaitelly, *Int J Infect Dis* 2023), clinically vulnerable individuals had a higher hazard ratio of death within 90 days following a primary infection (vs those without infection) compared with the less clinically vulnerable, and the survival benefit would be higher for the vulnerable vs less vulnerable beyond 90 days of the primary infection. In this study assessing the effectiveness of protection against (severe) reinfections, how does the healthy survivor effect affect the estimates for protection of prior infections with pre-Omicron/Omicron variants?

6. It was nice to see subgroup analyses were conducted to show the effectiveness of prior infection against reinfection by time since previous infection. In Table 4, it was claimed that the subgroup analyses in unvaccinated and vaccinated individuals showed similar results as the main analysis, but it would be interesting to discuss how the findings from this study can further shed light on immune responses to reinfections with different virus variants given the immunity from a primary infection or hybrid immunity, considering mechanisms like immune imprinting (Yisimayi, *Nature* 2023; Hornsby, *Nat Comm* 2023)?

Referee #4

(Remarks to the Author)

Thank you for the opportunity to review the manuscript, "Differential Patterns of Protection from Natural Infection against SARS-CoV-2 Reinfection in the Pre-Omicron and Omicron Eras". In this study, the authors used data from a country-level database of COVID-19-related data, including comprehensive testing and disease outcome information. This study database has been previously documented and described in the literature. The authors implemented a test-negative, case-control study design (as they have applied in previous studies) in order to measure the effectiveness of covid-19 infection against reinfection, for each of the pre-omicron and omicron variant periods. The authors find that for both pre-omicron and omicron eras, protection against severe outcomes remains highly durable, even a year after prior infection. However for post-omicron, compared to pre-omicron, protection against reinfection decays much faster and is mostly negligible after a year. The authors hypothesise that evolutionary pressure has driven immune escape post-omicron, compared to higher transmissibility being the main feature of viral mutation prior to the omicron era.

1. Overall the study is nicely contained, well-described and explained, and the methods seem sound. From a subjective perspective, I think this topic is of relevance. Certainly in my own field (dynamic modelling) estimates of the magnitude and durability of infection-induced protection from SARS-CoV-2 post-Omicron are lacking – there are not that many studies based on large-scale population data that quantify infection-induced protection estimates post-omicron. However in saying that I think it would be useful for the authors to help contextualise their results by drawing on and comparing with other estimates of infection protection that are in the literature (e.g. the results from the systematic review and meta-analysis in [https://doi.org/10.1016/S1473-3099\(22\)00801-5](https://doi.org/10.1016/S1473-3099(22)00801-5), or [https://doi.org/10.1016/S0140-6736\(22\)02465-5](https://doi.org/10.1016/S0140-6736(22)02465-5), or there may be more suitable studies for comparison), or explaining why it isn't appropriate to make such comparisons.

2. I did identify some issues that I hope the authors can fully explain and address. A key issue in this type of study is the bias introduced by misclassification of prior infection – in that prior infections, particularly if asymptomatic, may have been missed. There are a couple of points to make in the context of this study. I wonder if because of reduced propensity to test as we came out of the initial Omicron wave, it's more likely that infections would have been undetected in the post-Omicron part of the analysis – and what the impact would be on your results.

I note that this issue of bias is raised (only briefly) in the Discussion section, and you state that undocumented infections "may not have appreciably affected our estimates, as it has been demonstrated that even substantial misclassification of previous infection status had minimal impact on the estimated effectiveness of previous infection". However – the reference

cited here is a study by the same authors of this submitted manuscript (perhaps this should be pointed out in the text for full transparency). And upon having a closer look at that published reference, it seems that the misclassification of prior infection bias may actually have a large effect on estimates of protection in cases where >50% of the population has been previously infected – surely this would now be the case in the post-Omicron era? I think this potential bias needs to be given much more prominence, not just as a brief note in the discussion section, and the direction of bias should also be explained (and does it mean that the estimates of post-Omicron protection are potentially under-estimated?).

3. Testing patterns – rapid antigen tests were introduced just after the timing of Omicron emergence – so there was a known change in testing patterns. How was the variant determined for rapid antigen tests?

4. On the subgroup analysis considering unvaccinated and vaccinated individuals separately – I thought more could have been made of this part of the analysis? Did you consider including a visualisation of these results, and also outlining differences in results between the vaccinated and unvaccinated subgroups, and reasons for these?

5. Did you consider whether original antigenic sin could partly explain the more rapid waning of infection protection post-Omicron compared to the pre-Omicron era, rather than only the evolutionary pressure towards immune escape?

Version 1:

Reviewer comments:

Referee #1

(Remarks to the Author)

The revised manuscript has enhanced its scientific rigor, providing concise epidemiological data that underscore the distinct differences in protection efficiency against reinfection before and after the emergence of the Omicron variant.

However, it is still important to note that the conclusion that Omicron infections are less effective in preventing Omicron reinfections compared to the pre-Omicron era is not novel but a populational confirmation of prior molecular evidence.

Although I commend the significance of this evidence, the writings should be tuned down to that it's only an epidemiological confirmation. Besides, much molecular understanding, particularly of the nature and immunological impacts of SARS-CoV-2 ancestral strain immune imprinting, requires clarification and enhancement in the revised manuscript. Specific points are outlined below:

Starting from Line 110, the authors used the example of Beta strain to emphasize that molecular evidence can cause "unwarranted extrapolations". This is not appropriate. Indeed, Beta showed a high degree of immune evasion compared to the Wuhan strain, however, the fold-change of the NT50 titers of Beta and WT should not be used to correlate with protection efficacy directly; instead, the absolute neutralization titer is much more relevant. It's widely known that WT-vaccination-induced humoral immunity can neutralize Beta pretty well, despite the large fold-change observed, while Omicron BA.1 showed much lower absolute NT50 titers. Yes, people might not understand this well enough back in 2020-2021, but currently, it's well accepted that the antigenic distance between ancestral WT and Beta is not as significant as people thought, especially compared to the distance between Omicron BA.1 and later Omicron variants. This Beta example to emphasize the usefulness of populational evidence should be removed and frankly, it's not needed.

Short-term protection after exposure should not be used to assess any effects of immune imprinting. Actually, innate immunity plays a crucial role in short-term protection against reinfection, when the convalescent patients exhibit highly activated immunity against all pathogens. You could even observe short-term protection between influenza and COVID-19 infections in regional epidemiological data, where no cross-reactive immunity exists. Therefore, it is not persuasive to conclude any results regarding immune imprinting based solely on short-term protection data.

It is also essential to emphasize that immune imprinting does not imply a failure to induce Omicron-neutralizing antibodies. Rather, immune imprinting refers to the memory recall behavior of our immunity, in which the outcome depends on the antigenic distance between the ancestral strain and the variant. Simply speaking, Beta breakthrough infection would recall more neutralizing antibody-encoding memory B cells and induce higher neutralization titers against Beta, compared to the fact that Omicron BA.1 breakthrough infection would recall less neutralizing memory B and induce lower titers against BA.1 since BA.1 is more antigenic distinct to WT than Beta. This phenomenon actually fits very well to explain the observation of this paper.

Beyond neutralization titers or strengths, immune imprinting also influences the diversity of polyclonal neutralizing antibodies elicited by Omicron breakthrough infections. As Omicron evolves, it accumulates more evasive mutations and increasingly escapes more neutralizing antibodies. This would result in less diversity of the breakthrough infection-induced recalled neutralizing antibodies and concentrated selective pressure, which in turn accelerates its convergent evolution. This could also contribute to the observation of this paper.

In conclusion, it must be reiterated that the distinct protection efficiency observed in the pre-Omicron and Omicron eras is a highly complex outcome influenced by multiple interrelated factors. These factors include not only immune waning, evasion, and imprinting, but also the global population immunity faced by the strains, the accelerated and convergent evolution of Omicron, varying immunogenicity, and the population characteristics associated with infections at different stages of the pandemic. These factors are interconnected and challenging to disentangle. This study offers valuable phenomenological analyses that validate previous observations and experimental evidence. However, the explanatory and discussion sections are not convincing and require further refinement and tone down, and many mistakes should be revised.

Referee #4

(Remarks to the Author)

Thank you for the opportunity to provide an additional review of this study on the patterns of protection from natural infection relative to pre-omicron and post-omicron phases of the SARS-CoV-2 pandemic. I appreciated the authors' views and explanation about the potential role of immune imprinting in terms of the lower effectiveness of prior omicron infection against future omicron exposure – thank you for the thorough explanation and reasoning. I also appreciate the authors' thorough responses to my questions about study bias, particularly in relation to under-ascertainment of infections post-omicron and how this may have influenced the findings. The additional analyses that attempted to quantify the impact of this potential bias was welcome – it's important that this potential source of bias is highlighted sufficiently in the text. I am satisfied that the authors have addressed my main concerns.

A very minor point – but the resolution and the text font size of some of the supplementary figures (extended data figures 6 to 9) made them hard to read, so I suggest fixing this.

Referees' comments:

Referee #1 (Remarks to the Author):

This manuscript delivers important epidemiological and statistical analyses showing that pre-Omicron and Omicron infections exhibit distinct efficacies of protection against subsequent reinfections. Specifically, the protection induced by Omicron infection decreases significantly faster than the protection provided by pre-Omicron infection. The results underscore the importance of updating SARS-CoV-2 vaccines through clinical evidence, an aspect that should be appreciated and will interest those in this field, although this lacks discussion in the manuscript.

However, the accelerated evolution of Omicron lineages due to the immune pressure from neutralizing antibodies, highlighted by the authors as a potential explanation for the observation, has already been investigated at the molecular level and convincing evidence has been obtained in previous publications. And, the study lacks in-depth discussions of the potential relationship and interaction between the observed phenomenon, virus evolution, and the potential contribution of humoral immune imprinting. Despite its significance, I would not recommend this manuscript for publication in Nature.

Comment: We thank the reviewer for the time and effort put into this review, the assessment of our work, and the constructive feedback on our manuscript that enriched it and improved its readability. Please find below a point-by-point reply addressing each of the reviewer's comments.

Excellent point about the importance of updating SARS-CoV-2 vaccines, thank you. This point has now been added (Discussion, Page 17, Paragraph 1).

As for conceptual novelty, relationship to other lines of evidence, and immune imprinting as an explanation, please see our detailed responses below.

Major problem:

The major conclusion proposed by the author is already well-accepted in the field. Many virological analyses have obtained direct evidence regarding the difference in evolution tendency between the pre and post-Omicron eras

Answer: We appreciate the reviewer's perspective on conceptual novelty but respectfully disagree. While basic science research investigated viral evolution and immunity at the molecular level, these findings cannot necessarily be translated to actual population-level immune protection phenomenon. As the reviewer would appreciate, immune protection is a complex biological phenomenon that cannot be captured by a few lines of evidence in the laboratory. New vaccines, for example, must be investigated in trials on populations before their effectiveness in protecting against infection or severe disease can be quantified and established. During the pandemic, we have repeatedly observed that inferences from basic science data can lead to misleading conclusions when extrapolated to a population-level phenomenon.

For example, when the beta variant emerged, basic science evidence identified "extensive" mutations in the spike protein and demonstrated strong resistance (loss of activity by >10-fold) to neutralization by convalescent plasma and sera from vaccinated individuals.^{1,2} This led to the conclusion that this variant posed a major threat to the protective efficacy of the then newly developed vaccines and to natural immunity acquired through infection with the ancestral virus.^{1,2} However, these inferences proved to be unwarranted extrapolations. We have shown through direct population-level evidence that the functional effect of the beta mutations was minimal,³⁻⁵ offering much less loss in protection than expected based on basic science data. Our studies, which were the first to demonstrate these findings,³⁻⁵ were regarded as groundbreaking (thousands of citations in the scientific literature as well as extensive public interest and media coverage) because of the fundamentally novel findings they brought to our understanding of the impact of these specific viral mutations and their impact on immune protection.

This is not to diminish the critical importance of basic science data but rather to emphasize the essential role of population-level studies in providing conceptually novel findings with far-reaching implications. Similar to our beta studies, our present study provides the first direct population-level evidence contrasting the functional impact of viral evolution on immune protection in the pre-omicron versus the omicron era.

The implications of this study may also extend beyond SARS-CoV-2. Such population level evidence was not feasible before the COVID-19 pandemic, as it requires comprehensive and extensive data capture at a large (if not national) population level. The extensive resources devoted to the pandemic response, including widespread testing and documentation of infection and vaccination, made this study uniquely possible. While the extent to which these findings describe an ecological survival strategy for novel respiratory viruses capable of rapid mutation, such as RNA viruses, as they transition from emergence to endemicity remains unknown, similarities in evolution, transmission, and seasonality patterns across different respiratory infections⁶⁻⁸ suggest potential relevance to other viruses. This study may thus open new avenues for research, offering a more meaningful understanding of the mechanisms underlying the evolution of respiratory viruses.

The conceptual advancements and novelty of this study has now been indicated in the revised manuscript (Introduction, Page 5, Paragraph 3 and Page 6, Paragraph 1; and Discussion, Page 11, Paragraph 2).

Other factors may also contribute to the observation but lack discussions, such as:

1. It could result from immune imprinting. Due to the common ancestral strain vaccination history in the population, Omicron infection, which is antigenically distinct compared to the ancestral strain, fails to elicit sufficient Omicron-neutralizing antibodies.

Answer: Excellent point, thank you. We agree on the importance of immune imprinting as a biological phenomenon. In fact, we conducted a series of analyses to understand various aspects of it as it applies to SARS-CoV-2 natural infection and vaccine immunity.⁹⁻¹¹ However, the presented results do not support the idea that immune imprinting, and specifically the failure to elicit sufficient omicron-neutralizing antibodies, explains the findings for the following reasons:

- We observed similar levels of immune protection against reinfection right after infection in both the pre-omicron and omicron eras. The differences between the pre-omicron and omicron eras appeared only in the rate of waning (Fig. 1A). Both levels of immune protection started at approximately 80%, but waned at different rates. This suggests that omicron infection did not fail to elicit sufficient omicron-neutralizing antibodies; in fact, the initial protection was as strong as in the pre-omicron era. This is consistent with viral evolution and immune escape as a natural explanation, rather than immune imprinting.
- Confirming the above point, our earlier study conducted during and immediately after the first omicron wave investigated the short-term protection of BA.1 infection against BA.2 infection and vice versa.¹² We found that both omicron subvariants provided strong protection against each other in the first few weeks following infection.¹²
- In our two earlier studies investigating epidemiological evidence for immune imprinting related to both natural infection⁹ and vaccination,¹⁰ we found a consistent phenomenon with the same strong effect and effect size: omicron infection following pre-omicron immunity strengthens, rather than compromises, protection against subsequent omicron infection compared to omicron infection without prior pre-omicron immunity.^{9,10} Omicron infection after pre-omicron immunity strengthened the immune response against future reinfection, which is contrary to the idea of failing to elicit sufficient immune protection.^{9,10} This finding, once again, does not support immune imprinting as an explanation.
- The observed contrasting patterns between the pre-omicron and omicron eras were not only found in vaccinated individuals. The same patterns were observed in unvaccinated individuals. In the subgroup analyses for each group separately, we observed the same contrasting patterns despite the different immune histories (Table 4), further arguing against immune imprinting as an explanation.
- In the omicron era analysis, we excluded all individuals with a history of documented pre-omicron infection. While this does not completely exclude the possibility of undocumented infections, this exclusion should at least reduce the effect of the suggested immune imprinting if it were present. However, we observed clear and strongly contrasting patterns between the pre-omicron and omicron eras.

To address this comment, this point has now been discussed at length, along the lines presented above, in the Discussion section (Discussion, Page 11, Paragraphs 3 and 4 and Page 12, Paragraphs 1 and 2).

2. It could also be attributed to the rapid convergent evolution of the SARS-CoV-2 RBD.

Answer: Convergent evolution of the SARS-CoV-2 RBD has been observed in both the pre-omicron and omicron eras.^{13,14} Even if convergent evolution occurred more rapidly in one era compared to the other, this does not contradict our explanation of viral evolution and immune escape driving the observed contrasting patterns. Our study highlights a shift in evolutionary pressures, with intrinsic transmissibility driving adaptation in the pre-omicron era and immune escape becoming dominant in the omicron era. Convergent evolution is an inherent phenomenon of viral evolution itself that can facilitate both increased intrinsic transmissibility and enhanced immune escape. Moreover, immune imprinting can induce convergent SARS-CoV-2 RBD

evolution.¹⁵ Therefore, in our view, the available evidence discussed here on both convergent evolution and immune imprinting supports our explanation of viral evolution and immune escape driving the observed contrasting patterns.

We have now linked the issue of convergent evolution of the SARS-CoV-2 RBD to our findings in the Discussion section (Discussion, Page 10, Paragraph 3).

Despite I do enjoy the concise concept and results in this manuscript, I believe it would be more suitable to publish in a specialist journal.

Answer: We appreciate the reviewer's feedback and the intellectually rich discussion of these phenomena, which have enhanced this manuscript and its contribution to the literature. We hope that with the above responses and clarifications, we have demonstrated the conceptual novelty of this study, its advancement of knowledge of immune protection in two different pandemic eras, and the consistency of our explanation of viral evolution and immune escape driving the observed contrasting patterns.

Referee #3 (Remarks to the Author):

Chemaitelly et al. conducted a test-negative study investigating differential effectiveness particularly durability of previous infection with pre-Omicron SARS-CoV-2 variants in protection against pre-Omicron reinfections in comparison with protection of previous Omicron infection against Omicron reinfection using a population-based database of individuals who received a RT-PCR test for SARS-CoV-2 in the country during 5 Feb 2020-12 Feb 2024.

Comment: We thank the reviewer for the time and effort put into this review, the assessment of our work, and the constructive feedback on our manuscript that enriched it and improved its readability. Please find below a point-by-point reply addressing each of the reviewer's comments.

It's not surprising to see the study conclusion "These patterns suggest a shift in evolutionary pressures, with intrinsic transmissibility driving adaptation pre-omicron and immune escape becoming dominant post-omicron", which has already been shown in previous virologic and human studies (Carabelli, Nat Rev Microbiol 2023; Willet, Nat Rev Microbiol 2022), in addition to our general understanding of protection that tends to be long-lasting against infection with same/more similar strains.

Answer: We appreciate the reviewer's perspective on conceptual novelty but respectfully disagree. While basic science research investigated viral evolution and immunity at the molecular level, these findings cannot necessarily be translated to actual population-level immune protection phenomenon. As the reviewer would appreciate, immune protection is a complex biological phenomenon that cannot be captured by a few lines of evidence in the laboratory. New vaccines, for example, must be investigated in trials on populations before their effectiveness in protecting against infection or severe disease can be quantified and established.

During the pandemic, we have repeatedly observed that inferences from basic science data can lead to misleading conclusions when extrapolated to a population-level phenomenon. For example, when the beta variant emerged, basic science evidence identified "extensive" mutations in the spike protein and demonstrated strong resistance (loss of activity by >10-fold) to neutralization by convalescent plasma and sera from vaccinated individuals.^{1,2} This led to the conclusion that this variant posed a major threat to the protective efficacy of the then newly developed vaccines and to natural immunity acquired through infection with the ancestral virus.^{1,2} However, these inferences proved to be unwarranted extrapolations. We have shown through direct population-level evidence that the functional effect of the beta mutations was minimal,³⁻⁵ offering much less loss in protection than expected based on basic science data. Our studies, which were the first to demonstrate these findings,³⁻⁵ were regarded as groundbreaking (thousands of citations in the scientific literature as well as extensive public interest and media coverage) because of the fundamentally novel findings they brought to our understanding of the impact of these specific viral mutations and their impact on immune protection.

This is not to diminish the critical importance of basic science data but rather to emphasize the essential role of population-level studies in providing conceptually novel findings with far-reaching implications. Similar to our beta studies, our present study provides the first direct population-level evidence contrasting the functional impact of viral evolution on immune protection in the pre-omicron versus the omicron era.

The implications of this study may also extend beyond SARS-CoV-2. Such population level evidence was not feasible before the COVID-19 pandemic, as it requires comprehensive and extensive data capture at a large (if not national) population level. The extensive resources devoted to the pandemic response, including widespread testing and documentation of infection and vaccination, made this study uniquely possible. While the extent to which these findings describe an ecological survival strategy for novel respiratory viruses capable of rapid mutation, such as RNA viruses, as they transition from emergence to endemicity remains unknown, similarities in evolution, transmission, and seasonality patterns across different respiratory infections⁶⁻⁸ suggest potential relevance to other viruses. This study may thus open new avenues for research, offering a more meaningful understanding of the mechanisms underlying the evolution of respiratory viruses.

The conceptual advancements and novelty of this study has now been indicated in the revised manuscript (Introduction, Page 5, Paragraph 3 and Page 6, Paragraph 1; and Discussion, Page 11, Paragraph 2).

A series of studies have been published examining protection against reinfection/breakthrough infection with Omicron/pre-Omicron variants provided by previous infection/vaccination using the same database but perhaps from different periods. More technical details are needed to determine the validity of the study findings.

Answer: Yes, indeed, we have conducted various studies on immune protection from vaccination and natural infection. However, to our knowledge, this is the first study to provide direct population-level evidence of the contrasting patterns of immune protection between the pre-omicron and omicron eras. Please refer to our response to the previous comment regarding the

conceptual novelty of this study. Additionally, we have incorporated multiple revisions and additional analyses in the revised manuscript to address the reviewers' comments, providing more technical details and further validation of our findings.

A paragraph has been added to the Introduction section to provide context on existing work regarding protection against reinfection and to clarify the gap that the present study aims to fill (Introduction, Page 4, Paragraph 3). Additionally, please note the comparisons to existing evidence in the Discussion section (Discussion, Page 13, Paragraph 3).

1. Since the study period extends to Feb 2024 when the COVID-19 pandemic had been announced to end, I suspect the ascertainment of Omicron infections was substantially lower than that during the pre-Omicron period especially as explained in the Supplementary information that RATs were gradually increasing during Omicron era. How much would this affect the results of the analysis and therefore the interpretation of the findings?

Answer: Excellent point, thank you. The study used the test-negative design to generate the findings. Our previous research on this design demonstrated that even substantial under-ascertainment of infection (and thus misclassification of prior infection status) has minimal impact on estimating the effectiveness of infection against reinfection.¹⁶ This is, in fact, one of the key strengths of the test-negative design.¹⁶ Therefore, it is not likely that temporal differences in infection ascertainment could have impacted the findings of this study.

To further validate our results, the following analyses have been added to the revised manuscript:

- Mathematical modelling simulations were conducted by extending our earlier methodological work on the test-negative design¹⁶ to examine the impact of extreme under-ascertainment of infection (90% of infections are undocumented) on the pattern of waning immunity. This analysis was performed for both the pre-omicron and omicron patterns of waning immunity. The results demonstrated that the test-negative design remains robust in predicting patterns of waning immunity even under extreme conditions of infection under-ascertainment. This additional analysis is described in the Methods section (Methods, Page 22, Paragraph 1), its results are reported in the Results section (Results, Page 8, Paragraphs 4 and 5, and Page 9, Paragraph 1) and in a supplementary figure (Extended Data Fig. 5), and it is discussed in the Discussion section (Discussion, Page 14, Paragraph 1).
- The entire study was replicated using a different study design, specifically a cohort study design. The results from the cohort design confirmed the same contrasting patterns between the pre-omicron and omicron eras. This is notable because the test-negative and cohort study designs can theoretically be affected by different biases and to varying degrees. Despite this, both designs produced consistent findings. Importantly, in the cohort design, any temporal changes in infection ascertainment do not impact the estimated immune protection as long as these changes affect both cohorts non-differentially. There is no reason to believe these changes would affect each cohort differently. Additionally, we adjusted the cohort analyses for testing frequency to ensure that any differences in testing did not bias the results. This additional analysis is described in the Methods section (Methods, Page 23, Paragraphs 1-4; Page 24,

Paragraphs 1-4; and Page 25, Paragraph 1), its results are reported in the Results section (Results, Page 9, Paragraphs 4 and 5 and Page 10, Paragraph 1) and in two supplementary figures (Extended Data Figs. 8 and 9), and it is discussed in the Discussion section (Discussion, Page 16, Paragraphs 1 and 2).

2. Although the authors acknowledged that potential biases and confounders which might have not been accounted for sufficiently are the challenges faced in the real-world data analyses, it would be difficult to justify the validity of the analyses without further information on how much the current data might have been affected/ not affected by those potential biases such as non-binary healthcare seeking behaviors, testing behaviors/accessibility (to RT-PCR), timing of vaccination in addition to doses of vaccination, etc. As indicated in lines 201-208, the counterintuitive higher estimates of the effectiveness against any reinfection than that for symptomatic infections were explained by the differential timing between previous infection and reinfection. It would be more informative to have supporting data to verify how much the analyses would be affected by potential biases/confounders.

Answer: We agree with the reviewer that bias in real-world data can originate from various sources, including those mentioned by the reviewer. This is the reason our study was designed with comprehensive matching to mitigate potential biases. The analyses were conducted after matching for sex, age group, nationality, number of coexisting conditions, number of vaccine doses at the time of the SARS-CoV-2 test, calendar week of the SARS-CoV-2 test, testing method, and reason for testing. Consequently, the factors highlighted by the reviewer have been accounted for in our analyses. Furthermore, this matching strategy has been validated in previous studies with different epidemiologic designs, employing control groups to assess null effects.^{5,17-20} These studies have corroborated that our matching approach effectively neutralizes differences in infection exposure.^{5,17-20}

The example noted by the reviewer of a counterintuitive estimate for symptomatic infection is not a consequence of bias but results from testing in waves versus times of low incidence (different distributions of routine versus symptoms-based testing). The median time between the previous infection and the study test was 245 days for the analysis of any reinfection, while it was longer at 301 days for the analysis of symptomatic reinfection. This longer interval in the latter analysis resulted in greater waning of immune protection, leading to the observed difference in effectiveness estimates. This also has no bearing on the contrasting patterns of immune protection between the pre-omicron and omicron eras, which is the focus of our study. The same contrasting patterns were observed for both any infection and symptomatic infection (Tables 2 and 3, respectively).

To further validate our results, please note the additional analyses of mathematical modelling simulations and the replication of the entire study using a different study design (cohort study design), as described in our response to the previous comment. These analyses, which address different sources of bias, confirmed our results and findings. The first analysis is described in the Methods section (Methods, Page 22, Paragraph 1), its results are reported in the Results section (Results, Page 8, Paragraphs 4 and 5, and Page 9, Paragraph 1) and in a supplementary figure (Extended Data Fig. 5), and it is discussed in the Discussion section (Discussion, Page 14,

Paragraph 1). The second analysis is described in the Methods section (Methods, Page 23, Paragraphs 1-4; Page 24, Paragraphs 1-4; and Page 25, Paragraph 1), its results are reported in the Results section (Results, Page 9, Paragraphs 4 and 5 and Page 10, Paragraph 1) and in two supplementary figures (Extended Data Figs. 8 and 9), and it is discussed in the Discussion section (Discussion, Page 16, Paragraphs 1 and 2).

We have now also added the timing between the last vaccine dose and the SARS-CoV-2 test for cases and controls, which showed similar durations for cases and controls in both the pre-omicron and omicron analyses (Table 1). Please also note the other revisions and additional analyses described below in response to other comments and the detailed discussion of the matching in the Methods section (Methods, Page 19, Paragraph 3 and Page 20, Paragraph 1) and Discussion section (Discussion, Page 15, Paragraphs 2 and 3 and Page 16, Paragraphs 1 and 2).

3. Supplementary Table 1, why is the overall estimate of the effectiveness of pre-Omicron infection against reinfection of pre-Omicron variants lower than any of the estimates from the subgroup analyses, a similar question for the estimate for <1 year in subgroup analysis 2?

Answer: This is a subtle point. This table reports a sensitivity analysis in which a 40-day time window was used to define reinfection instead of the conventional 90-day time window (Extended Data Table 1). During much of the pre-omicron phase of the pandemic, a negative PCR test was required for some activities, leading to extensive testing. Since PCR is highly sensitive and can detect the virus at low viral loads, including non-viable fragments, and because there were rare cases where the infection lasted more than 40 days,²¹ a small number of positive PCR tests were classified as reinfections based on the 40-day criteria, even though they were not true reinfections. These cases biased the protection estimates to slightly lower values.

This bias, however, impacted only the estimates that included SARS-CoV-2 tests between 40 days and 90 days. This is why the slight underestimation influenced the estimates for only overall protection and <1 year protection, as indicated by the reviewer, but did not affect the other estimates.

This point has now been clarified in the Discussion section (Discussion, Page 14, Paragraph 3).

4. As described, comorbidities were defined based on the records in the public healthcare system. How much do the authors think the under-ascertainment of information on those whose comorbidities were not captured by the database would affect their results? Any supporting data for that?

Answer: We do not believe that this has consequences on the reported contrasting patterns between the pre-omicron and omicron eras. The influence of comorbidity on the risk of infection (not severe disease) is expected to be minimal. To validate this point, a sensitivity analysis was conducted for the revised manuscript, in which matching by comorbidities was removed to simulate a scenario of complete under-ascertainment. This analysis confirmed the same results.

This additional analysis is described in the Methods section (Methods, Page 22, Paragraph 2), its results are reported in the Results section (Results, Page 9, Paragraphs 2 and 3) and in two supplementary figures (Extended Data Figs. 6 and 7), and it is discussed in the Discussion section (Discussion, Page 16, Paragraphs 1 and 2).

5. As they previously showed (Chemaitelly, Int J Infect Dis 2023), clinically vulnerable individuals had a higher hazard ratio of death within 90 days following a primary infection (vs those without infection) compared with the less clinically vulnerable, and the survival benefit would be higher for the vulnerable vs less vulnerable beyond 90 days of the primary infection. In this study assessing the effectiveness of protection against (severe) reinfections, how does the healthy survivor effect affect the estimates for protection of prior infections with pre-Omicron/Omicron variants?

Answer: Indeed, depletion of the primary-infection cohort within the population due to COVID-19 mortality at the time of primary infection can skew this cohort towards healthier individuals.²² This skew could potentially lead to an overestimation of the effectiveness of prior infection in preventing severe, critical, or fatal COVID-19 upon reinfection. Nevertheless, given the low COVID-19 mortality rate in Qatar's predominantly young population (less than 0.1% of primary infections),²²⁻²⁵ a survival effect is not likely to appreciably alter the observed very high effectiveness of prior infection in mitigating the severity of reinfection.

This point has now been discussed in the Discussion section (Discussion, Page 14, Paragraph 2).

6. It was nice to see subgroup analyses were conducted to show the effectiveness of prior infection against reinfection by time since previous infection. In Table 4, it was claimed that the subgroup analyses in unvaccinated and vaccinated individuals showed similar results as the main analysis, but it would be interesting to discuss how the findings from this study can further shed light on immune responses to reinfections with different virus variants given the immunity from a primary infection or hybrid immunity, considering mechanisms like immune imprinting (Yisimayi, Nature 2023; Hornsby, Nat Comm 2023)?

Answer: Excellent point and useful references, thank you. We agree on the importance of immune imprinting as a biological phenomenon and its links to different forms of hybrid immunity. In fact, we conducted a series of analyses to understand various aspects of it as it applies to SARS-CoV-2 natural infection and vaccine immunity.⁹⁻¹¹ To address this comment, three paragraphs have now been added to the Discussion section to discuss immune imprinting and its potential explanation of the findings of this study (Discussion, Page 11, Paragraphs 3 and 4 and Page 12, Paragraphs 1 and 2). We have also added a paragraph specifically discussing the results of vaccinated versus unvaccinated individuals and linking them to immune imprinting effects, including the cited references (Discussion, Page 13, Paragraph 2).

Referee #4 (Remarks to the Author):

Thank you for the opportunity to review the manuscript, "Differential Patterns of Protection from Natural Infection against SARS-CoV-2 Reinfection in the Pre-Omicron and Omicron Eras". In this study, the authors used data from a country-level database of COVID-19-

related data, including comprehensive testing and disease outcome information. This study database has been previously documented and described in the literature. The authors implemented a test-negative, case-control study design (as they have applied in previous studies) in order to measure the effectiveness of covid-19 infection against reinfection, for each of the pre-omicron and omicron variant periods. The authors find that for both pre-omicron and omicron eras, protection against severe outcomes remains highly durable, even a year after prior infection. However for post-omicron, compared to pre-omicron, protection against reinfection decays much faster and is mostly negligible after a year. The authors hypothesise that evolutionary pressure has driven immune escape post-omicron, compared to higher transmissibility being the main feature of viral mutation prior to the omicron era.

Comment: We thank the reviewer for the time and effort put into this review, the assessment of our work, and the constructive feedback on our manuscript that enriched it and improved its readability. Please find below a point-by-point reply addressing each of the reviewer's comments.

1. Overall the study is nicely contained, well-described and explained, and the methods seem sound. From a subjective perspective, I think this topic is of relevance. Certainly in my own field (dynamic modelling) estimates of the magnitude and durability of infection-induced protection from SARS-CoV-2 post-Omicron are lacking – there are not that many studies based on large-scale population data that quantify infection-induced protection estimates post-omicron. However in saying that I think it would be useful for the authors to help contextualise their results by drawing on and comparing with other estimates of infection protection that are in the literature (e.g. the results from the systematic review and meta-analysis in [https://doi.org/10.1016/S1473-3099\(22\)00801-5](https://doi.org/10.1016/S1473-3099(22)00801-5), or [https://doi.org/10.1016/S0140-6736\(22\)02465-5](https://doi.org/10.1016/S0140-6736(22)02465-5), or there may be more suitable studies for comparison), or explaining why it isn't appropriate to make such comparisons.

Answer: Thank you for the useful suggestions. A paragraph has been added to the Introduction section to provide context on existing work regarding protection against reinfection and to clarify the gap that the present study aims to fill (**Introduction, Page 4, Paragraph 3**). Additionally, please note the comparisons to existing evidence in the Discussion section (**Discussion, Page 13, Paragraph 3**).

2. I did identify some issues that I hope the authors can fully explain and address. A key issue in this type of study is the bias introduced by misclassification of prior infection – in that prior infections, particularly if asymptomatic, may have been missed. There are a couple of points to make in the context of this study. I wonder if because of reduced propensity to test as we came out of the initial Omicron wave, it's more likely that infections would have been undetected in the post-Omicron part of the analysis – and what the impact would be on your results.

I note that this issue of bias is raised (only briefly) in the Discussion section, and you state that undocumented infections “may not have appreciably affected our estimates, as it has been demonstrated that even substantial misclassification of previous infection status had minimal impact on the estimated effectiveness of previous infection”. However – the

reference cited here is a study by the same authors of this submitted manuscript (perhaps this should be pointed out in the text for full transparency). And upon having a closer look at that published reference, it seems that the misclassification of prior infection bias may actually have a large effect on estimates of protection in cases where >50% of the population has been previously infected – surely this would now be the case in the post-Omicron era? I think this potential bias needs to be given much more prominence, not just as a brief note in the discussion section, and the direction of bias should also explained (and does it mean that the estimates of post-Omicron protection are potentially under-estimated?).

Answer: Excellent and subtle point, thank you. The reviewer raises a valid point regarding potential misclassification bias of prior infection status in the test-negative design, particularly when a large proportion of the population (>50%) is immune.¹⁶ However, in our study, this is less of a concern due to the waning immunity observed over time. While the majority of the population has already been infected in the post-omicron era, the level of immunity varies across individuals due to the rapid waning of this protection. This gradual decline in protection mitigates the risk of reaching a scenario where a very high proportion of the population is immune, which would render the test-negative design susceptible to seriously underestimating immune protection.

To demonstrate this point and validate our results, the following analyses have been added to the revised manuscript:

- Mathematical modelling simulations were conducted by extending our cited work on the test-negative design¹⁶ to assess the impact of extreme under-ascertainment of infections (where 90% of infections are undocumented) on the pattern of waning immunity. This analysis was performed for both the pre-omicron and omicron periods. The results showed that the estimated effectiveness in the presence of this bias, during both the pre-omicron and omicron phases, was lower but still comparable to the true effectiveness (Extended Data Fig. 5). Importantly, this bias did not affect the estimated pattern and duration of immune protection, which is the central focus of our study. Notably, this simulation exaggerates the impact of this bias. The simulation results are reported at the 2-year mark from the onset of both the pre-omicron and omicron pandemic phases, which is at the end of these phases. However, in the actual study reported in this manuscript, the tests are conducted throughout these phases, at times when population immunity is lower than at the end of these phases, resulting in less impact from this bias. This additional analysis is described in the Methods section (Methods, Page 22, Paragraph 1), its results are reported in the Results section (Results, Page 8, Paragraphs 4 and 5, and Page 9, Paragraph 1) and in a supplementary figure (Extended Data Fig. 5), and it is discussed in the Discussion section (Discussion, Page 14, Paragraph 1).
- The entire study was replicated using a different study design, specifically a cohort study design. The results from the cohort design confirmed the same contrasting patterns between the pre-omicron and omicron eras. This is notable because the test-negative and cohort study designs can theoretically be affected by different biases and to varying degrees. Despite this, both designs produced consistent findings. This additional analysis is described in the Methods section (Methods, Page 23, Paragraphs 1-4; Page 24, Paragraphs 1-4; and Page 25, Paragraph 1), its results are reported in the Results section

(Results, Page 9, Paragraphs 4 and 5 and Page 10, Paragraph 1) and in two supplementary figures (Extended Data Figs. 8 and 9), and it is discussed in the Discussion section (Discussion, Page 16, Paragraphs 1 and 2).

We have also clarified in the text that the earlier work investigating the test-negative design is ours, as suggested by the reviewer (Discussion, Page 14, Paragraph 1).

3. Testing patterns – rapid antigen tests were introduced just after the timing of Omicron emergence – so there was a known change in testing patterns. How was the variant determined for rapid antigen tests?

Answer: Rapid antigen testing was introduced to alleviate the pressure on PCR testing during the massive first omicron wave (Extended Data Fig. 3). All analyses were conducted after matching for the calendar week of the SARS-CoV-2 test, testing method (PCR or rapid antigen), and reason for testing. This was done to meet the requirement of the test-negative design, which mandates that cases and controls exhibit the same healthcare-seeking behaviour.^{16,26,27} Consequently, the changes in testing method during the pandemic were controlled for in this study design. This clarification has now been added to the Methods section (Methods, Page 19, Paragraph 3 and Page 20, Paragraph 1).

The variant status was not determined for each individual SARS-CoV-2 test but was set based on the dominant variant at the time of infection diagnosis. Infections were categorized as pre-omicron or omicron. The duration of dominance for every variant throughout the pandemic was determined using Qatar's variant genomic surveillance,²⁸⁻³⁰ which includes viral genome sequencing²⁸ and multiplex real-time reverse-transcription PCR (RT-qPCR) variant screening²⁹ of weekly collected random positive clinical samples. Once the first massive omicron wave started, virtually all infections were due to omicron, as it displaced all pre-omicron variants.³¹⁻³³ This clarification has now been added to the Methods section (Methods, Page 20, Paragraph 3).

4. On the subgroup analysis considering unvaccinated and vaccinated individuals separately – I thought more could have been made of this part of the analysis? Did you consider including a visualisation of these results, and also outlining differences in results between the vaccinated and unvaccinated subgroups, and reasons for these?

Answer: Thank you for the useful suggestions. We have now added a visualization of these results as suggested (Extended Data Fig. 4). We have also commented on them in the opening paragraph of the Discussion section (Discussion, Page 10, Paragraph 2) and discussed them in a specific paragraph for these specific results (Discussion, Page 13, Paragraph 2). Please also note our responses to Referees #1 and #2 in relation to immune imprinting and the added point related to these specific results (Discussion, Page 12, Paragraph 2).

5. Did you consider whether original antigenic sin could partly explain the more rapid waning of infection protection post-Omicron compared to the pre-Omicron era, rather than only the evolutionary pressure towards immune escape?

Answer: Excellent point, thank you. We agree on the importance of original antigenic sin (immune imprinting) as a biological phenomenon. In fact, we conducted a series of analyses to understand various aspects of it as it applies to SARS-CoV-2 natural infection and vaccine immunity.⁹⁻¹¹ To address this comment, we have now added three paragraphs to the discussion of our findings (**Discussion, Page 11, Paragraphs 3 and 4 and Page 12, Paragraphs 1 and 2**). These provide an argument against immune imprinting as an explanation for these specific observed patterns, and support viral evolution and immune escape as a natural explanation. Additionally, please refer to our response to Comment #1 by Referee #1, which closely relates to the point raised here.

References

- 1 Wang, P. *et al.* Antibody resistance of SARS-CoV-2 variants B.1.351 and B.1.1.7. *Nature* **593**, 130-135, doi:10.1038/s41586-021-03398-2 (2021).
- 2 Planas, D. *et al.* Sensitivity of infectious SARS-CoV-2 B.1.1.7 and B.1.351 variants to neutralizing antibodies. *Nat Med* **27**, 917-924, doi:10.1038/s41591-021-01318-5 (2021).
- 3 Abu-Raddad, L. J., Chemaitelly, H., Butt, A. A. & National Study Group for Covid Vaccination. Effectiveness of the BNT162b2 Covid-19 Vaccine against the B.1.1.7 and B.1.351 Variants. *N Engl J Med* **385**, 187-189, doi:10.1056/NEJMc2104974 (2021).
- 4 Chemaitelly, H., Bertollini, R., Abu-Raddad, L. J. & National Study Group for Covid Epidemiology. Efficacy of Natural Immunity against SARS-CoV-2 Reinfection with the Beta Variant. *N Engl J Med* **385**, 2585-2586, doi:10.1056/NEJMc2110300 (2021).
- 5 Chemaitelly, H. *et al.* mRNA-1273 COVID-19 vaccine effectiveness against the B.1.1.7 and B.1.351 variants and severe COVID-19 disease in Qatar. *Nat Med* **27**, 1614-1621, doi:10.1038/s41591-021-01446-y (2021).
- 6 Ferguson, N. M., Galvani, A. P. & Bush, R. M. Ecological and immunological determinants of influenza evolution. *Nature* **422**, 428-433, doi:10.1038/nature01509 (2003).
- 7 Kissler, S. M., Tedijanto, C., Goldstein, E., Grad, Y. H. & Lipsitch, M. Projecting the transmission dynamics of SARS-CoV-2 through the postpandemic period. *Science* **368**, 860-868, doi:10.1126/science.abb5793 (2020).
- 8 Lavine, J. S., Bjornstad, O. N. & Antia, R. Immunological characteristics govern the transition of COVID-19 to endemicity. *Science* **371**, 741-745, doi:10.1126/science.abe6522 (2021).
- 9 Chemaitelly, H. *et al.* Immune Imprinting and Protection against Repeat Reinfection with SARS-CoV-2. *N Engl J Med* **387**, 1716-1718, doi:10.1056/NEJMc2211055 (2022).
- 10 Chemaitelly, H. *et al.* History of primary-series and booster vaccination and protection against Omicron reinfection. *Sci Adv* **9**, eadh0761, doi:10.1126/sciadv.adh0761 (2023).
- 11 Chemaitelly, H. *et al.* Long-term COVID-19 booster effectiveness by infection history and clinical vulnerability and immune imprinting: a retrospective population-based cohort study. *Lancet Infect Dis* **23**, 816-827, doi:10.1016/S1473-3099(23)00058-0 (2023).
- 12 Chemaitelly, H. *et al.* Protection of Omicron sub-lineage infection against reinfection with another Omicron sub-lineage. *Nat Commun* **13**, 4675, doi:10.1038/s41467-022-32363-4 (2022).
- 13 Bouhaddou, M. *et al.* SARS-CoV-2 variants evolve convergent strategies to remodel the host response. *Cell* **186**, 4597-4614 e4526, doi:10.1016/j.cell.2023.08.026 (2023).
- 14 Roemer, C. *et al.* SARS-CoV-2 evolution in the Omicron era. *Nat Microbiol* **8**, 1952-1959, doi:10.1038/s41564-023-01504-w (2023).
- 15 Cao, Y. *et al.* Imprinted SARS-CoV-2 humoral immunity induces convergent Omicron RBD evolution. *Nature* **614**, 521-529, doi:10.1038/s41586-022-05644-7 (2023).
- 16 Ayoub, H. H. *et al.* Estimating protection afforded by prior infection in preventing reinfection: Applying the test-negative study design. *Am J Epidemiol*, doi:10.1093/aje/kwad239 (2023).
- 17 Abu-Raddad, L. J. *et al.* Pfizer-BioNTech mRNA BNT162b2 Covid-19 vaccine protection against variants of concern after one versus two doses. *J Travel Med* **28**, doi:10.1093/jtm/taab083 (2021).

- 18 Chemaitelly, H. *et al.* Waning of BNT162b2 Vaccine Protection against SARS-CoV-2 Infection in Qatar. *N Engl J Med* **385**, e83, doi:10.1056/NEJMoa2114114 (2021).
- 19 Abu-Raddad, L. J., Chemaitelly, H., Bertollini, R. & National Study Group for Covid Vaccination. Waning mRNA-1273 Vaccine Effectiveness against SARS-CoV-2 Infection in Qatar. *N Engl J Med* **386**, 1091-1093, doi:10.1056/NEJMc2119432 (2022).
- 20 Abu-Raddad, L. J., Chemaitelly, H., Bertollini, R. & National Study Group for Covid Vaccination. Effectiveness of mRNA-1273 and BNT162b2 Vaccines in Qatar. *N Engl J Med* **386**, 799-800, doi:10.1056/NEJMc2117933 (2022).
- 21 Abu-Raddad, L. J. *et al.* Two prolonged viremic SARS-CoV-2 infections with conserved viral genome for two months. *Infect Genet Evol* **88**, 104684, doi:10.1016/j.meegid.2020.104684 (2021).
- 22 Chemaitelly, H. *et al.* Short- and longer-term all-cause mortality among SARS-CoV-2-infected individuals and the pull-forward phenomenon in Qatar: a national cohort study. *Int J Infect Dis* **136**, 81-90, doi:10.1016/j.ijid.2023.09.005 (2023).
- 23 Seedat, S. *et al.* SARS-CoV-2 infection hospitalization, severity, criticality, and fatality rates in Qatar. *Sci Rep* **11**, 18182, doi:10.1038/s41598-021-97606-8 (2021).
- 24 Chemaitelly, H. *et al.* Turning point in COVID-19 severity and fatality during the pandemic: a national cohort study in Qatar. *BMJ Public Health* **1**, e000479, doi:10.1136/bmjph-2023-000479 (2023).
- 25 AlNuaimi, A. A. *et al.* All-cause and COVID-19 mortality in Qatar during the COVID-19 pandemic. *BMJ Glob Health* **8**, doi:10.1136/bmjgh-2023-012291 (2023).
- 26 Jackson, M. L. & Nelson, J. C. The test-negative design for estimating influenza vaccine effectiveness. *Vaccine* **31**, 2165-2168, doi:10.1016/j.vaccine.2013.02.053 (2013).
- 27 Verani, J. R. *et al.* Case-control vaccine effectiveness studies: Preparation, design, and enrollment of cases and controls. *Vaccine* **35**, 3295-3302, doi:10.1016/j.vaccine.2017.04.037 (2017).
- 28 Benslimane, F. M. *et al.* One Year of SARS-CoV-2: Genomic Characterization of COVID-19 Outbreak in Qatar. *Front Cell Infect Microbiol* **11**, 768883, doi:10.3389/fcimb.2021.768883 (2021).
- 29 Hasan, M. R. *et al.* Real-Time SARS-CoV-2 Genotyping by High-Throughput Multiplex PCR Reveals the Epidemiology of the Variants of Concern in Qatar. *Int J Infect Dis* **112**, 52-54, doi:10.1016/j.ijid.2021.09.006 (2021).
- 30 Saththasivam, J. *et al.* COVID-19 (SARS-CoV-2) outbreak monitoring using wastewater-based epidemiology in Qatar. *Sci Total Environ* **774**, 145608, doi:10.1016/j.scitotenv.2021.145608 (2021).
- 31 Abu-Raddad, L. J. *et al.* Effect of mRNA Vaccine Boosters against SARS-CoV-2 Omicron Infection in Qatar. *N Engl J Med* **386**, 1804-1816, doi:10.1056/NEJMoa2200797 (2022).
- 32 Altarawneh, H. N. *et al.* Effects of Previous Infection and Vaccination on Symptomatic Omicron Infections. *N Engl J Med* **387**, 21-34, doi:10.1056/NEJMoa2203965 (2022).
- 33 Altarawneh, H. N. *et al.* Protection against the Omicron Variant from Previous SARS-CoV-2 Infection. *N Engl J Med* **386**, 1288-1290, doi:10.1056/NEJMc2200133 (2022).

Differential Protection Against SARS-CoV-2 Reinfection Pre- and Post-Omicron

REPLY TO REVIEWERS' COMMENTS

We are grateful to the reviewers for assessing our work and for their insightful and useful feedback and suggestions. Please find below a point-by-point reply addressing each of the comments. We have also incorporated these suggestions in the revised manuscript, as noted below. We would be pleased to address any additional matters, should that be necessary.

Note: All references to the revised manuscript pertain to the marked copies of the manuscript files including changes implemented through "track changes".

Reviewers' comments:

Referee #1 (Remarks to the Author):

The revised manuscript has enhanced its scientific rigor, providing concise epidemiological data that underscore the distinct differences in protection efficiency against reinfection before and after the emergence of the Omicron variant.

Comment: We thank the reviewer for the time and effort put into this review, the assessment of our work, and the constructive feedback on our manuscript that enriched it and improved its readability. Please find below a point-by-point reply addressing each of the reviewer's comments.

However, it is still important to note that the conclusion that Omicron infections are less effective in preventing Omicron reinfections compared to the pre-Omicron era is not novel but a populational confirmation of prior molecular evidence. Although I commend the significance of this evidence, the writings should be tuned down to that it's only an epidemiological confirmation. Besides, much molecular understanding, particularly of the nature and immunological impacts of SARS-CoV-2 ancestral strain immune imprinting, requires clarification and enhancement in the revised manuscript. Specific points are outlined below:

Answer: We greatly appreciate the reviewer's insightful and useful feedback, which has enriched this article and strengthened its contribution to the literature. As outlined in our responses to the specific points below, we have highlighted that this study provides population-level results that validate previous experimental molecular evidence (Discussion, Page 10, Paragraph 2). Additionally, we have substantially revised the discussion of the contributing factors explaining the observed contrasting patterns, incorporating the reviewer's suggestions and insights (Discussion, Page 10, Paragraphs 2-3 and Page 11, Paragraphs 1-2).

Starting from Line 110, the authors used the example of Beta strain to emphasize that molecular evidence can cause "unwarranted extrapolations". This is not appropriate. Indeed, Beta showed a high degree of immune evasion compared to the Wuhan strain, however, the fold-change of the NT50 titers of Beta and WT should not be used to correlate with protection efficacy directly; instead, the absolute neutralization titer is much more relevant. It's widely known that WT-vaccination-induced humoral immunity can neutralize Beta pretty well, despite the large fold-change observed, while Omicron BA.1 showed much lower absolute NT50 titers. Yes, people might not understand this well enough back in 2020-2021, but currently, it's well accepted that the antigenic distance between ancestral WT and Beta is not as significant as people thought, especially compared to the distance between Omicron BA.1 and later Omicron variants. This Beta example to emphasize the usefulness of populational evidence should be removed and frankly, it's not needed.

Answer: We agree with the reviewer and apologize for the confusion. The beta variant example was intended to emphasize the reviewer's point that some molecular data "should not be used to correlate directly with protection efficacy." However, as the reviewer noted, many people misinterpreted the initial molecular evidence following the emergence of the beta variant in 2020-2021, leading to unwarranted conclusions. This reinforces the critical role of epidemiological evidence in corroborating molecular findings at the population level.

To prevent confusion, we have removed the discussion of the beta variant as suggested (Introduction, Page 5, Paragraph 3).

Short-term protection after exposure should not be used to assess any effects of immune imprinting. Actually, innate immunity plays a crucial role in short-term protection against reinfection, when the convalescent patients exhibit highly activated immunity against all pathogens. You could even observe short-term protection between influenza and COVID-19 infections in regional epidemiological data, where no cross-reactive immunity exists. Therefore, it is not persuasive to conclude any results regarding immune imprinting based solely on short-term protection data.

It is also essential to emphasize that immune imprinting does not imply a failure to induce Omicron-neutralizing antibodies. Rather, immune imprinting refers to the memory recall behavior of our immunity, in which the outcome depends on the antigenic distance between the ancestral strain and the variant. Simply speaking, Beta breakthrough infection would recall more neutralizing antibody-encoding memory B cells and induce higher neutralization titers against Beta, compared to the fact that Omicron BA.1 breakthrough infection would recall less neutralizing memory B and induce lower titers against BA.1 since BA.1 is more antigenic distinct to WT than Beta. This phenomenon actually fits very well to explain the observation of this paper.

Beyond neutralization titers or strengths, immune imprinting also influences the diversity of polyclonal neutralizing antibodies elicited by Omicron breakthrough infections. As Omicron evolves, it accumulates more evasive mutations and increasingly escapes more neutralizing antibodies. This would result in less diversity of the breakthrough infection-induced recalled

neutralizing antibodies and concentrated selective pressure, which in turn accelerates its convergent evolution. This could also contribute to the observation of this paper.

In conclusion, it must be reiterated that the distinct protection efficiency observed in the pre-Omicron and Omicron eras is a highly complex outcome influenced by multiple interrelated factors. These factors include not only immune waning, evasion, and imprinting, but also the global population immunity faced by the strains, the accelerated and convergent evolution of Omicron, varying immunogenicity, and the population characteristics associated with infections at different stages of the pandemic. These factors are interconnected and challenging to disentangle. This study offers valuable phenomenological analyses that validate previous observations and experimental evidence. However, the explanatory and discussion sections are not convincing and require further refinement and tone down, and many mistakes should be revised.

Answer: Since the above comments are related, we are responding to them together. First, we greatly appreciate the reviewer's insightful and informative feedback. We agree that the two distinct patterns observed in the omicron versus pre-omicron eras are likely the result of a complex interplay of multiple interrelated factors, in addition to waning immunity, and that it is prudent to avoid overinterpreting the findings. Instead, the focus should remain on what the results provide—a description of a population-level phenomenon.

Accordingly, we have revised the Discussion section by: 1) highlighting that the observed distinct patterns are likely a complex outcome influenced by multiple interrelated factors in addition to waning immunity; 2) including a brief discussion of these other potential factors; 3) revising the discussion of immune imprinting by incorporating the reviewer's insights and shortening it to avoid drawing definitive conclusions based solely on the observed patterns; and 4) toning down several statements that extend beyond the immediate findings of the study (Discussion, Page 10, Paragraphs 2-3 and Page 11, Paragraphs 1-2).

Referee #4 (Remarks to the Author):

Thank you for the opportunity to provide an additional review of this study on the patterns of protection from natural infection relative to pre-omicron and post-omicron phases of the SARS-CoV-2 pandemic. I appreciated the authors' views and explanation about the potential role of immune imprinting in terms of the lower effectiveness of prior omicron infection against future omicron exposure – thank you for the thorough explanation and reasoning. I also appreciate the authors' thorough responses to my questions about study bias, particularly in relation to under-ascertainment of infections post-omicron and how this may have influenced the findings. The additional analyses that attempted to quantify the impact of this potential bias was welcome – it's important that this potential source of bias is highlighted sufficiently in the text. I am satisfied that the authors have addressed my main concerns.

Comment: We thank the reviewer for the time and effort put into this review, the assessment of our work, and the constructive feedback on our manuscript that enriched it and improved its

readability. Please find below a point-by-point reply addressing each of the reviewer's comments.

A very minor point – but the resolution and the text font size of some of the supplementary figures (extended data figures 6 to 9) made them hard to read, so I suggest fixing this.

Answer: All extended data figures have been revised to conform with the Journal's guidelines (Extended Data Figs. 1-9).